# CODING RELIABLE LLM-BASED INTEGRATED TASK AND KNOWLEDGE AGENTS WITH GENIEWORKSHEETS

## ABSTRACT

Large Language Models (LLMs) present an opportunity to create automated assistants that can help users navigate complex tasks. However, existing approaches have limitations in handling conditional logic, integrating knowledge sources, and consistently following instructions. Researchers and industry professionals often employ ad hoc pipelines to construct conversational agents. These pipelines aim to maintain context, address failure cases, and minimize hallucinations, yet frequently fail to achieve these objectives. To this end, we present Genie – a programmable framework for creating task-oriented conversational agents that are designed to handle complex user interactions and knowledge queries. Unlike LLMs, Genie provides reliable grounded responses, with controllable agent policies through its expressive specification, Genie Worksheet. In contrast to dialog trees, it is resilient to diverse user queries, helpful with knowledge sources, and offers ease of programming policies through its declarative paradigm. The agents built using Genie outperforms the state-of-the-art method on complex logic domains in STARV2 dataset by up to 20.5%. Through a real-user study involving 62 participants, we show that Genie beats the GPT-4 Turbo with function calling baseline by 21.1%, 20.1%, and 61% on execution accuracy, dialogue act accuracy, and goal completion rate, respectively, on three diverse real-world domains.

## 1 INTRODUCTION

Researchers and industry practitioners have demonstrated significant interest in developing task-oriented dialogue agents. These agents are typically designed with a transactional focus, aiming to fill slot values based on user utterances to complete specific tasks (Budzianowski et al., 2018; Andreas et al., 2020; Rastogi et al., 2020). However, natural conversations between users and call agents cannot be adequately modeled solely as slot-filling tasks. Users often require external information to progress through the steps necessary for task completion. For example, when attempting to make a restaurant reservation, users frequently seek to identify a venue that best aligns with their preferences. Additionally, users may interrupt the agent at any point during the conversation to pose new inquiries. Traditional dialogue tree-based approaches often struggle to handle such unexpected user questions or deviations, leading to a suboptimal user experience (Bocklisch et al., 2017; Xie et al., 2022; Amazon, 2023; Press, 2024; Google, 2024).

The advent of Large Language Models (LLMs) offers a promising opportunity to create more natural dialogue agents. These LLM-based agents can effectively navigate "unhappy paths," where the system adeptly manages unforeseen user inquiries at any stage of the conversation. Furthermore, LLMs have enabled the development of knowledge assistants capable of querying structured data by generating query languages such as SQL (Pourreza & Rafiei, 2023b) and SPARQL (Liu et al., 2024c) and retrieving text for unstructured knowledge (Khattab et al., 2023). However, LLMs can be unreliable, posing challenges for developers aiming to deploy them in real-world applications, primarily due to a lack of control over conversations and inaccurate contextualization of information.

**Challenge 1: Creation of effective, informative, and responsive informative agents, while letting developers exercise control without onerous efforts.** Developers desire to maintain control over critical aspects of the agent's operation, including the flow of conversations, the timing of actions, and the information elicited from users. For example, if a user declines an agent's offer to book a restaurant, developers should be able to program the agent to suggest alternative options, such as

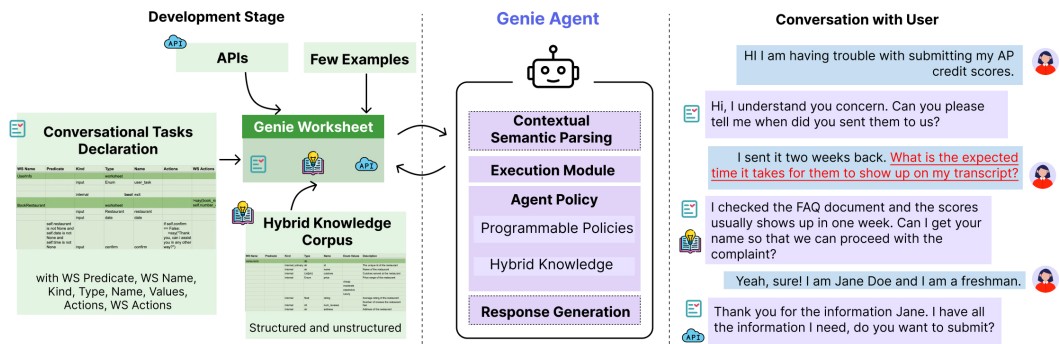

Figure 1: Overview of the Genie Assistant. (Left) The Genie Worksheet supplied by the developer consists of specifications of conversational tasks, the schema of the knowledge corpus, APIs to complete the tasks, and a few examples for the LLM parser. The tasks are represented in a spreadsheet containing the variables pertinent to the dialogue and actions to take based on their values. (Middle) The Genie Worksheet is used by the Genie framework to create a dialogue agent. (Right) The resulting agent can reliably carry out many accurate conversations while requiring little developer effort.

offering discounts. With the in-context learning capabilities of LLMs, researchers have begun incorporating agent policies as instructions in model prompts (Zhang et al., 2023b; Liu et al., 2024a). However, LLMs often fail to adhere strictly to these instructions, resulting in inconsistent and unreliable agent behavior (Liu et al., 2024b). Recent libraries, such as those offered by LangChain and Guidance, provide abstractions for developing LLM-based agents but still require developers to manually craft prompts and create reliable pipelines.

**Challenge 2. Support users' queries for information, which may be embedded in a task request.** Semantic parsing, which maps natural language to logical forms, has been applied to API invocations (Patil et al., 2023) and database queries (Pourreza & Rafiei, 2023a; Liu et al., 2024d). To create task agents that can answer questions, Kim et al. (2020) proposes using intent classification to generate a query or an API invocation. However, in practice, a single user statement can combine a query with an API invocation. For example, "I want to book a romantic restaurant in London on Valentine's day", translates into a query to produce a restaurant to be used as input to a reservation API invocation. How do we express and support arbitrary compositions of queries and API calls?

**Challenge 3: Dialogue systems need to remember pertinent facts from the dialogue history.** Current methods based on LLMs typically rely on including in the prompt a complete dialogue history (Ulmer et al., 2024; Liu et al., 2024a) or a summary of the conversation (Packer et al., 2023; Li et al., 2024a). However, full dialogue histories can cause LLMs to overlook critical details in extended interactions; summarization may omit essential information, leading to repetitive questioning or generating inaccurate responses, often referred to as "hallucinations" (Bang et al., 2023).

**Our proposed method**. This paper proposes a method that let developers easily create dependable and effective task-oriented agents that can carry out accurate, fluid, and informative conversations with users. We address the above challenges with a novel *high-level*, *expressive* and *declarative* specification representation we have created called a Genie Worksheet. The core concept is to ask the developer to directly declare variables to represent information that is pertinent to the conversation and to describe the actions to perform according to the input information. For answering user questions, the developer only have to supply the knowledge bases, which may be structured or unstructured. In other words, the developer is only declaring the *necessary* specification, and we show that the specification is *sufficient* to create effective knowledge and task-oriented agents.

From the Genie Worksheet, the Genie framework derives a conversational agent that fills in the variables in the Worksheet as the conversation continues. *The actively updated Worksheets double up as a succinct representation of the dialogue state*, designed carefully to address the complexity in semantic parsing and agent policies. As shown in Figure 1, an agent processes each turn of the conversation in four steps: a contextual neural semantic parser to interpret the input, an execution module performs the queries and actions, an algorithmic agent policy module decides on the agent actions, and a neural response generator generates the response governed by the agent policy.

Contributions of this paper include:

1. Genie is the first work that generates a robust integrated knowledge and task-oriented agent from a declarative specification using a novel high-level specification language (Genie Worksheet).

   • It is the first representation that supports *full integration* of knowledge and task-oriented agents. Unlike previous slot-based task-oriented dialogues (ToD), the agent can handle utterances that contain formal knowledge queries and whose results are used in API calls **(Challenge 2)**.

   • It is a high-level *declarative* language that lets the developer *control the agent policy* easily. Developers only need to specify (1) spreadsheets for tasks that associate actions with the requested information from the user and task completion, (2) schemas of structured databases and free-text repositories **(Challenge 1)**.

   • Unlike ToD agents, which support only a fixed set of parameters, Genie Worksheet is *expressive*, as it supports dialogues where the information to request is *dependent on users' answers*. Such behavior, typical of call agents, has previously been supported only in dialog trees.

   • Genie Worksheet facilitates the derivation of an effective agent, as it is instrumental in the creation of an accurate semantic parser and a flexible agent policy. It is the representation used as *the context and target for the semantic parser*, allowing the parser to remember pertinent information in a long conversation **(Challenge 3)**. It is also interpreted by the agent policy module, which can handle the complexity of *mixed-initiative* dialogues easily by first servicing the user's initiatives before the agent's initiatives as recorded in the worksheets **(Challenge 1)**, as shown in the second to last turn of the agent in Figure 1.

2. Using a fully functional prototype of Genie Worksheet, Genie agents outperform the SOTA on StarV2 (Zhao et al., 2022) by up to 20.5% points across the complex logic domains.

3. Via a user study of 62 participants across 3 different real-life dialogue-oriented tasks, we show that Genie agents are significantly better than the previous SOTA. The results show an impressive 86.5% execution accuracy, 89.2% dialog act accuracy, and an 82.8% goal completion rate, surpassing the GPT-4 function-calling baseline by 21.1, 20.1, and 61 percentage points, respectively.

## 2 DESIGN AND RATIONALE OF THE GENIE WORKSHEET

The objective in the design of the worksheet is to provide developers a high-level, concise, expressive specification of the task at hand. Our worksheet design is inspired by the versatility of webforms. Modern websites contain multiple fields which can be optional, tabs for selection of task, and pop-up windows that depend on previous user responses. To facilitate task completion and access to knowledge bases, Genie Worksheet allows two kinds of worksheets: a *task* worksheet and a knowledge base worksheet, described below.

**Task Worksheet**  Users perform different tasks based on their requirements. For example, a student who is having trouble enrolling in a class would need to fill out a form containing details about the course they want to enroll in, the error message they are seeing, and any other additional comments and tasks like Leave of Absence are irrelevant to them. Hence, to make only relevant worksheets available to the user, all the worksheets have a *WS Predicate* field indicating when a task worksheet should be activated, as illustrated in Figure 2. The task worksheet also has a *WS Name* and a *WS Kind* set to Task. Additionally, to provide flexibility to the agent, Genie Worksheet allows executing arbitrary Python code to call external APIs, change assigned values, or explicitly respond to the user with a given string using built-in actions like `say`. The actions are triggered when all the required parameters for a task are assigned and are defined under *WS Actions*.

Each worksheet contains a set of fields. A field has a *Predicate*, similar to the *WS Predicate*, it indicates when a field can be filled in. Genie allows three kinds of fields. The user provides values for `input` fields; the `output` fields are set according to the output of executing an API or knowledge base call; and the `internal` fields can only be manipulated by the agent. These are defined under *Kind*. The *Description* field provides a natural language description of the field. Each field is tied to a *Type*, where all conventional types are allowed. If it is an enum *Type* (e.g., student_task in Figure 2), a set of *Enum Values* are specified in the succeeding lines. The boolean *DontAsk* will record the information if the user offers it, but the agent will not ask for it. If DontAsk is false, the agent will ask for the field if it is not assigned, but user can refuse to answer if it is not *Required*. Whenever a field is assigned, similar to *WS Actions*, the supplied *Actions* should be executed either

| | Ticket Submission Worksheet | | | | | | | | | | | |
|---|---|---|---|---|---|---|---|---|---|---|---|---|
| WS Predicate | WS Name | Predicate | Kind | Type | Name | Enum Values | Description | Don't Ask | Required | Confirmation | Actions | WS Actions |
| | Main | | Task | | | | | | | | | >say(submit_ticket(self.student_task, ...)) |
| | | | input | Enum | student_task | | The type of student requ... | | TRUE | | | |
| | | | | | | TroubleShoot | | | | | | |
| | | | | | | Leave of Abs. | | | | | | |
| | | | | | | Test Credits | | | | | | |
| | | self.student_task == "TroubleShoot" | input | Trouble Shoot | trouble_shoot | | The enrollment issues that the student is facing | | TRUE | | | |
| | | | | | | ••• | | | | | | |
| | | self.student_task is not None and ( self.trouble_shoot and self.leave_of_abs and self.test_credits) | input | str | extra_details | | Ask for any other detail that the student wants to add | | TRUE | | | |
| | | | input | confirm | confirm | | Confirm that the student wants to submit the ticket. | | TRUE | | | if self.confirm == False: >say("Thank you, how else can I help you?") > exitws() |
| self.student_task == "Trouble..." | TroubleShoot | | WS | | | | | | | | | |
| | | | | | | ••• | | | | | | |
| | services_general_info | | KB | | | | | | | | | |

Figure 2: The Genie Worksheet specification for the support ticket submission assistant in Figure 1.

if the *Confirmation* boolean is set to false or if the user has confirmed with the agent. We provide additional details on the specification in Appendix A.

**Knowledge Access** Real-life user utterances can contain task statements and knowledge queries. Hence, to handle such utterances, Genie Worksheet treats knowledge access as a first-class object. Real-life queries often involve both structured and unstructured accesses. For instance, queries "What's the best-rated restaurant with a romantic atmosphere" require access to both the structured "ratings" column and the free text "reviews" column. To handle hybrid knowledge bases, Genie adopts the SUQL query language, an SQL extension that integrates search of unstructured data (Liu et al., 2024d). For each knowledge base to be included, the developer must create a worksheet with *Kind* set to KB. The fields in the KB Worksheet define the schema of the KB.

# 3 GENIE: RELIABLE LLM-BASED DIALOGUE AGENTS

## 3.1 FORMAL REPRESENTATION

Current LLM-based dialogue agents contextualize the conversation by providing the complete dialogue history or summarizing it. However, despite their impressive natural understanding capabilities, they often struggle with maintaining relevant contextual information across longer conversations. Additionally, summarizing conversations can cause a loss of key information in the conversation. This leads to hallucinated or fabricated values and repeating questions. Therefore, to mitigate these issues, Genie tracks the dialogue state with a formal dialogue state.

A dialogue consists of a set of alternating user and agent turns, $\{u_1, a_1, \cdots, u_{t-1}, a_{t-1}, u_t\}$, where $u_i \in \mathcal{U}$, $a_i \in \mathcal{A}$ denoting user and agent utterances.

**Dialogue State** As an agent runs, it assigns values to the fields in task worksheets, executes actions, and performs queries. Figure 3 shows the conversation history in the course enrollment domain, where the student's goal is to enroll in courses for the next semester. We represent dialogue at each turn $t$, $d_t \in D$, as a sequence of records, $r \in R$, each of which can be a completed or partially specified task or a KB query. For example, $r_1$ in Dialogue State ($d_{t-1}$) corresponds to the knowledge query $u_{t-1}$, where the student asks for courses where students are happy with the workload. Each record in the dialogue state has a *kind* of either task or KB query, an *initiator*, which can be an agent or user. As shown in $r_1$, a KB query record has four additional attributes: (1) *names*, the set of names of knowledge bases mentioned in the query (*courses* and *ratings* in $r_1$), (2) *NL query*: a string of the form `answer(s)`, where `s` is the de-contextualized (i.e. self-contained) query in natural language, (3) *KB query*: the formal query associated with the NL query, (4) *result*: the result of executing the formal query. A task record has two additional attributes: (1) *name*, the name of a worksheet, (2) *assignment*, which maps one or more fields in the worksheet to its value. For example, $r_2$ in $d_{t-1}$ corresponds to the task of choosing a course. Hence, the kind is *Task*, and the name is the *Course* worksheet. The compositions of queries and APIs are supported by passing the result of an instance of a record in as a field of another record. The assignment of `course_id` in record $r_2$ is $r_1$, which finds the courses where students are happy with the course load.

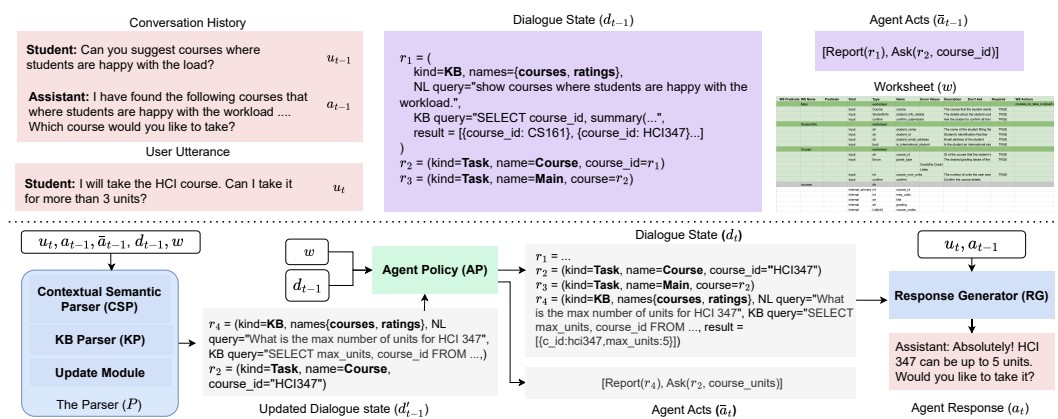

Figure 3: Genie provides the latest set of worksheets and only one previous conversation turn to the semantic parser. The parser outputs the current worksheets, which are used by the Agent Policy to generate the Agent Dialogue Acts. These Acts along with the latest worksheet value and the latest user utterance are used to generate the response.

**Agent Acts** Agent utterances have a formal representation ($\bar{a} \in \bar{A}$) consisting of 5 agent acts:

- REPORT (record): Report the result obtained by executing the task or query in the record.
- CONFIRM (record, fld name): Confirm the value mapped to a field name in a record.
- SAY (utterance): Explicitly say the given utterance.
- PROPOSE (record): Propose to the user a new task or query, with possibly partially pre-filled values, in the given record.
- ASK (record, fld name): Ask for the value of a field in a record.

In Figure 3, the agent acts $\bar{a}_{t-1}$ contains the REPORT act that corresponds to the assistant responding to the student's query ($u_{t-1}$) with courses that where student are happy with the workload in the conversation history ($a_{t-1}$). The ASK corresponds to the assistant asking for which course the student would want to enroll in $a_{t-1}$.

## 3.2 ARCHITECTURE OF THE GENIE AGENT

The Genie agent has three components:

- $P(u_t, a_{t-1}, \bar{a}_{t-1}, d_{t-1}, w) : \mathcal{U} \times \mathcal{A} \times \mathcal{P}(\bar{A}) \times \mathcal{D} \times \mathcal{W} \to \mathcal{D}$: A parser $P$ accepts the current user utterance $u_t$ and contextualizes the utterance using the last agent utterance $a_{t-1}$, the previous-turn agent acts $\bar{a}_{t-1}$, the formal dialogue state from the last turn $d_{t-1}$, worksheet $w$, and generate a new dialogue state $d$.
- $AP(d, d', w) : \mathcal{D} \times \mathcal{D} \times \mathcal{W} \to \mathcal{P}(\bar{A}) \times \mathcal{D}$: an agent policy that accepts the dialogue state from the previous turn $d$, the new dialogue state after parsing $d'$, and the worksheet $w$ and produces a set of agent dialogue acts $\bar{a}$, along with a new dialogue state.
- $RG(d_t, \bar{a}_t, u_t, a_{t-1}) : \mathcal{D} \times \mathcal{P}(\bar{A}) \times \mathcal{U} \times \mathcal{A} \to \mathcal{A}$: a response generator that accepts the formal dialogue state $d_t$, an agent act set $\bar{a}_t$, the last user $u_t$ utterance and agent utterance $a_{t-1}$ to generate the agent response $a_t$.

Figure 3 shows the overview the Genie system. At turn $t$ in the conversation, the system is given: (1) the conversation history $\{u_1, a_1, \cdots, u_{t-1}, a_{t-1}, u_t\}$, where $u_t \in \mathcal{U}$, $a_t \in \mathcal{A}$ denoting user and agent utterances; (2) the $t-1$-th turn formal agent act $\bar{a}_{t-1} \in \mathcal{P}(\bar{A})$; (3) the $n-1$-th turn formal dialogue state $d_{t-1} \in \mathcal{D}$; and (4) the Genie Worksheet $w \in \mathcal{W}$.

**The Parser** consists of three modules. The contextual semantic parsing (CSP) that translates the user utterance into a set of changes to be applied to the dialogue state. The context supplied to CSP is a compressed version of the dialogue state, where only the last KB record is included. The user may (1) *supply values* to fields that are anticipated in existing worksheets; (2) *modify* a previously

---

**Algorithm 1** Genie Agent Policy

---

1: **Input:** Previous dialogue state $d$, updated dialogue state from the parser $d'$, worksheet: $w$
2: **Output:** New dialogue state $d'$, Agent Act $\bar{a}$
3: Initialize a new Agent Acts set $\bar{a} \leftarrow$ empty set
4: **if** KB record $r$ exists in $\mathrm{Diff}(d', d)$ **then**
5:     **if** required parameter $p$ associated with $r$ is unfilled **then**
6:         $\bar{a} \leftarrow \bar{a} \cup \{\textsc{Ask}(r, p)\}$                         ▷ Ask for the first required parameter
7:     **else**
8:         Execute the KB query in the KB record $r$
9:         $\bar{a} \leftarrow \bar{a} \cup \{\textsc{Report}(r)\}$                    ▷ Report the result of KB record to the user
10:     **end if**
11: **end if**
12: **for all** assigned fields $f$ in worksheets present in $\mathrm{Diff}(d', d)$ **do**
13:     **if** $f$ requires confirmation **then**
14:         $\bar{a} \leftarrow \bar{a} \cup \{\textsc{Confirm}(r, f)\}$                      ▷ Ask for confirmation
15:     **else**
16:         Execute the actions associated with the field $f$
17:         Update the state $d'$ and add new agent acts to $\bar{a}$
18:     **end if**
19: **end for**
20: **for all** Task records $r$ that are complete but whose actions have not been executed **do**
21:     Execute the actions associated with the worksheet for $r$
22:     Update the state $d'$ and add new agent acts to $\bar{a}$
23: **end for**
24: **for all** required fields $f$ in the records $r$ in $d'$ **do**
25:     **if** $f$ has not been assigned **then**
26:         $\bar{a} \leftarrow \bar{a} \cup \{\textsc{Ask}(r, f)\}$ **break**                  ▷ Ask for the first unfilled field
27:     **end if**
28: **end for**
29: **Return** updated dialogue state $d'$ and $\bar{a}$

---

filled field or remove its value; (3) *initiate* new tasks or queries. If the user enters values into fields or modifies previously filled fields, the queries are update statements that modify an existing record, as illustrated in Figure 4. If the user initiates a new task or query, a new record is created. The CSP leaves queries to knowledge bases in natural language.

The second module, the knowledge parser (KP), translates natural language queries into formal queries. This function decomposition allows for more advanced methods for knowledge retrieval (Yao et al., 2022; 2023) and abstracts knowledge integration away from CSP. The third module performs the updates on the dialogue state to create an updated dialogue state. The implementation details and prompts are in Appendix G.1.

As in Figure 3, given the user utterance ($u_t$), *"I will take the HCI course. Can I take it for more than 3 units?"*, the CSP assigns the course "HCI347" to the record $r_2$, and the KB Parser generates a KB query $r_1$ to find how many units the student can take for the HCI course.

**Agent Policy** As opposed to relying on LLM to directly generate agent response based on conversation so far, Genie uses a symbolic module to compute the agent responses. The rationale for this is twofold: (1) Providing agent acts causes the LLM to generate a deterministic response governed by

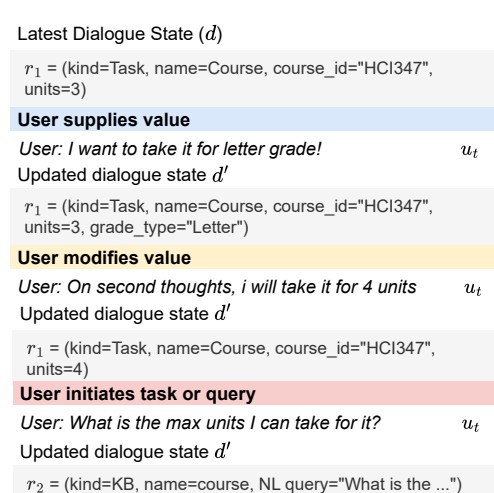

Figure 4: The Parser P produces updated dialogue state $d'$ when the user supplies value or modifies them, and creates new instances when the user initiates a new task or query.

the agent policy. (2) LLMs struggle with under-represented developer-defined policies and cannot

follow all the instructions. Using a symbolic module to compute the necessary actions improves the LLM's ability to follow instructions.

The agent policy (AP) interprets the updated dialogue state $d' \in \mathcal{D}$ generated by the parser ($P$), previous dialogue state $d$ and the worksheet signature $w \in \mathcal{W}$ to inform its actions, resulting in an update of the dialogue state $d' \in \mathcal{D}$ and a set of new agent acts $\bar{a} \in \mathcal{P}(\bar{\mathcal{A}})$. The agent policy (AP) utilizes the `Diff`$(d', d)$ function that identifies any fields and worksheets that have been modified or initiated between two dialogue states $d'$ and $d$.

Algorithm 1 outlines the agent policy. The procedure initiates by assessing whether a new KB record exists within the updated dialogue state $d'$ compared to $d$ (Line 4). If any required parameters $p$ are missing in the record, the agent prompts the user to provide the missing values (Line 6). Otherwise, the system reports the result of the KB query (Line 9). Subsequently, the policy checks whether any new fields require confirmation and prompts users (Line 14). If no confirmation is needed, actions associated with the newly filled fields are executed (Line 16), and the dialogue state is updated accordingly (Line 17). Any agent acts based on developer defined actions are recorded (Line 18). If all fields in any worksheet are filled and their corresponding actions have been completed, the policy executes the worksheet's actions (Line 22) and updates the dialogue state (Line 23). Any agent acts based on actions are recorded (Line 24). Finally, the policy identifies the first unfilled field in the dialogue state and generates an agent act requesting the user to provide the missing value (Line 28).

As in our example in Figure 3, the policy first executes the KB query in $r_4$ and adds the result of the KB query as REPORT ($r_4$) act to the agent acts list. It then checks all the required fields in record $r_2$ and finds that the value of field "course_units" is unfilled and hence adds ASK ($r_2$, course_units).

**Response Generation** Once we have the final dialogue state $d'$ and the agent's dialogue acts $\bar{a}$ produced by AP, we generate a response to the user using an LLM. The LLM takes in the previous user and agent utterance $(u, a)$ as additional context. The LLM is instructed to adhere to the agent acts $\bar{a}$ as opposed to making up its own actions. As shown in Figure 3, the agent response ($a_t$) "... HCI 347 can be up to 5 units. Would you like to take it?" corresponds to the REPORT and ASK agent acts ($\bar{a}_t$) generated by the agent policy. The prompt is shown in Appendix H.1.

## 3.3 DISCUSSION

Genie automatically handles the dialgoue flow based on a succinct declarative Worksheet specification. This is significantly different from dialogue trees which require the programmer to hand-code the policy for each turn. Another popular approach is to model the conversation as a finite state machine where user inputs are mapped to a small number of user dialogue acts and rules are used to map the user dialogue acts, together with results of API or DB queries, to agent dialogue acts. As we discussed, each turn can invoke multiple different agent dialogue acts, resulting in an exponentially many agent dialogue state, had a finite machine been used. Purely function calling in LLMs will also fail to handle the many details in a conversation. In Appendix B, we show some conversational snippets that we can handle well but can cause other techniques to fail, along with the reasons.

## 4 EVALUATION ON STATIC DATASETS

We perform a comparison study on the complex logic domains in a slot-based benchmark – STARv2 dataset (Zhao et al., 2022), where existing agents perform poorly.

**Baselines:** We compare Genie to AnyTOD (Zhao et al., 2022) the SOTA result on StarV2. AnyTOD finetunes T5 XXL (13B) model on all domains except the one tested (roughly 6000 data points). (-SGD) refers to additional fine-tuning on Schema Guided Dialogue. (-PROG) denotes the use of programmable policies. We also evaluate current SOTA LLMs: Llama 3.1 Instruct 70B (Dubey et al., 2024) and GPT-4 Turbo. We prompt the LLM with natural language policy (an example is shown below), current belief state (dialogue state) as provided in STARv2, the agent acts with description, and the user utterance. We ask the LLM to select the next agent act according to the given input. We provide experimental details in the Appendix D.

**Slot filling performance:** We observe that a zeroshot GPT-4 Turbo-based agent is competitive against a finetuned AnyTOD (AT-SGD XXL) and outperforms it in the Bank and Trivia domain. However, agents with programmable policies on top of AnyTOD outperform zeroshot

agents. Genie based on GPT-4 turbo outperforms all the baselines across all domains, beating the previous SOTA (AT-PROG+SGD XXL) by up to 20.5%. This validates that Genie is generalizable to the slot-filling paradigm. Furthermore, our analysis indicates that most of our errors are caused by inconsistent data annotations, also highlighted by (Zhao et al., 2022).

**Can Genie improve performance of base LLM?** In Table 2, we evaluate how different base LLM affect the performance of Genie using Llama 3.1 Instruct 70B and GPT 4-turbo. We observe that Genie significantly improves the performance of the base models Llama and GPT-4 Turbo across all domains by up to 33.16 points and 40.70 points, respectively. The experiments also highlight that in some domains, weaker models (llama 3.1 70b) can achieve similar performance to much stronger models (GPT-4 turbo). For example, in the banking domain, Genie with Llama falls short of GPT-4 Turbo by only 0.4 points.

| Agent | Bank | Trip | Trivia |
|---|---|---|---|
| *Finetuned T5 (11B)* | | | |
| AT XXL | 54.3 | 52.4 | 73.8 |
| AT-SGD XXL | 53.1 | 51.5 | 81.1 |
| AT-PROG XXL | 61.0 | 60.8 | 73.7 |
| AT-PROG +SGD XXL | 65.0 | 62.9 | 86.3 |
| *Zeroshot* | | | |
| Llama 3.1 70B | 48.9 | 41.7 | 81.7 |
| GPT-4 Turbo | 55.1 | 42.7 | 82.5 |
| Genie (Ours) | **82.5** | **83.4** | **92.7** |

Table 1: System Action F1 for complex logic domains in STAR V2.

| *Llama 3.1-70B* | | | | *GPT 4 (turbo)* | | | |
|---|---|---|---|---|---|---|---|
| **Agent** | **Bank** | **Trip** | **Trivia** | **Agent** | **Bank** | **Trip** | **Trivia** |
| Base LLM | 48.9 | 41.7 | 81.7 | Base LLM | 55.1 | 42.7 | 82.5 |
| Genie (Base LLM) | 82.1 | 75.9 | 82.22 | Genie (Base LLM) | 82.5 | 83.4 | 92.7 |

Table 2: Genie improves base LLMs significantly up to 40.7 points on STAR V2.

## 5 EVALUATION ON REAL USER STUDIES

### 5.1 APPLICATIONS

We choose three diverse applications with varying complexities. Compared to the 3 StarV2 domains, these real-world applications are much more complicated, as shown in Table 5 in the Appendix.

**Restaurant Reservation** Making a restaurant reservation requires finding a suitable restaurant and providing booking information to complete a transaction. We use the real-life dataset containing restaurants from Yelp.com from Liu et al. (2024d).

**Ticket Submission** In this study, we aim to replicate a subset of tasks found within a university's student services portal. University student services portals typically contain various tasks categorized under different sections and subsections, posing a navigational challenge for students seeking to locate the appropriate link. Moreover, these portals often contain a vast corpus of free-text data, which students must peruse before submitting a ticket. We evaluate agents' capability to handle nested webpages with predicates and subsequent actions.

**Course Enrollment** Finally, we evaluate the performance of Genie as a course enrollment assistant, which combines hybrid data sources to search for course details and fill out complicated nested forms. The assistants allow students to ask questions about course requirements, student reviews, and ratings while filling out their enrollment forms. We collect a real-life dataset containing courses from the Computer Science program, with 4 tables (courses, offerings, ratings, and programs).

### 5.2 EXPERIMENTAL SETTING

**Baselines:** We compare our system against GPT-4-turbo with functional calling, which we call GPT-4 (FC). We provide the baseline system with the ability to use the same KB Parser for external knowledge access. This baseline closely follows Li et al. (2024b). We also experimented with Nemo-Guardrails (Rebedea et al., 2023), a recently-proposed development framework for LLM-based conversational agents, but found it fails miserably on basic task-oriented dialogues (refer to Appendix F for details). All Genie Worksheets are provided in Appendix K.

| | All Domains | | Restaurant | | Ticket Submission | | Course Enrollment | |
|---|---|---|---|---|---|---|---|---|
| | GPT4 (FC) | Genie | GPT4 (FC) | Genie | GPT4 (FC) | Genie | GPT4 (FC) | Genie |
| SP Acc | - | 91.4 | - | 93.8 | - | 85.8 | - | 94.1 |
| Ex Acc | 65.4 | 86.5* | 50.0 | 88* | 58.6 | 80.0* | 79.2 | 89.7* |
| DA Acc | 69.1 | 89.2* | 57.7 | 92.5* | 67.6 | 82.5* | 77.8 | 92.9* |
| Goal CR | 21.8 | 82.8* | 54.5 | 91.6 | 0.0 | 80.0* | 10.0 | 80.0* |

Table 3: We perform a real user study with 62 users across three domains. We find that Genie performs significantly better than GPT-4 with Function Calling ability on the three applicable metrics. We perform t-test and mark the results with $p < 0.05$ with *. Underline represents the better result.

**Study Design:** We use Prolific to recruit 22 participants for Restaurant Reservations and 20 users who identified as students for Ticket Submission. We recruited 20 university students to evaluate the Course Enrollment assistant. We instructed them to attempt to book a reservation, submit an issue ticket, and enroll in two courses, respectively. We randomly assigned users to one of the two visually identical systems. We collected 99, 81, and 127 turns with Genie, and 90, 70, and 144 turns with With GPT 4 (FC), respectively. More details on user instructions are in Appendix I.

**Evaluation Metrics:** We use four evaluation metrics and manually inspect the conversation to assess each metric. (1) Semantic Parsing Accuracy (SP Acc) is used to evaluate if the natural language user utterance corresponds to the correct APIs, databases, and filled fields. (2) Execution Accuracy (Ex Acc) to check whether the agent executes the correct API and databases. (3) Agent Dialogue Act Accuracy (DA Acc) checks if the agent follows the policies provided by the developer. (4) Goal Completion Rate (Goal CR) to evaluate the user's ability to successfully complete the task with the system's assistance. We provide additional details in Appendix D.1.

## 5.3 MAIN RESULTS

Table 3 compares Genie against GPT 4 (FC) across three metrics and finds that Genie performs significantly better than GPT-4 (FC) on all three domains. We observe that Genie consistently demonstrates a high semantic parsing rate (exceeding $85\%$), indicating that Genie Worksheet is simple to understand for LLMs with few shot examples. The marginally lower SP Acc ($85.8\%$) observed in the Ticket Submission application can be attributed to the application's complexity, featuring several worksheets (=8) and fields (=28), as shown in Table 5 in the Appendix.

Additionally, the higher Ex Acc can be attributed to providing compressed context as the formal dialogue state, enabling the LLM to invoke the correct API and execute the suitable knowledge query. The superior performance on DA Acc can be ascribed to our agent's capability to provide turn-by-turn instructions rather than presenting all the instructions at once, as is the case with GPT-4 (FC). **The results validate the benefit of programmable policies in delivering a reliable assistant**.

In terms of the final Goal Completion Rate, we observe that GPT-4 (FC) scores relatively higher in restaurant reservations, the domain it is most familiar with and well represented in existing dialogue datasets (Ye et al., 2022; Rastogi et al., 2020). Similar results are also observed by Zhang et al. (2023b) and Hudeček & Dusek (2023). However, GPT 4 (FC) struggles with less familiar domains, like ticket submission and course enrollment. We observe that the higher number of predicates, necessitating several instructions in the Ticket Submission application, makes it extremely challenging for GPT-4 (FC) to assist users in completing their tasks. In most cases, GPT-4 (FC) does not elicit all the required fields from the user, making incomplete API calls. Moreover, despite having access to the KB Parser, **GPT-4 (FC) often hallucinates** non-existent courses and fails to enroll students.

## 5.4 ERROR ANALYSIS

We analyze the mean dialogue act accuracy (DA Acc) as a function of the fraction of turns in the conversation for both Genie and GPT-4 (FC) in Figure 6. We observe that Genie experiences a comparatively smaller performance drop, approximately 8%, compared to GPT-4's 43%. As conversations lengthen, this performance degradation becomes more pronounced. This supports our hypothesis that LLMs struggle to maintain context over extended interactions while employing a formal dialogue state mitigates some of these issues. We also analyze all the error cases for our system and GPT-4 (FC) and provide examples in the Appendix.

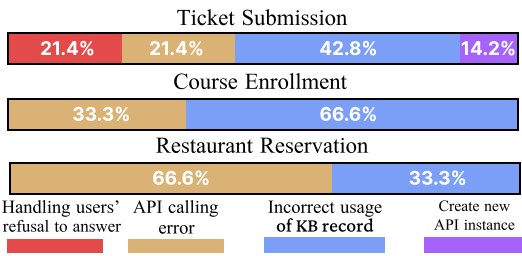
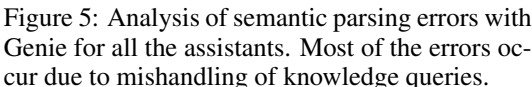

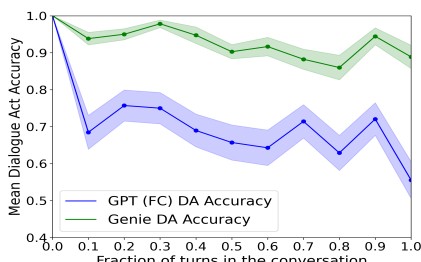

Figure 5: Analysis of semantic parsing errors with Genie for all the assistants. Most of the errors occur due to mishandling of knowledge queries.

Figure 6: The mean dialogue act accuracy for GPT-4 (FC) and Genie as a function of the conversation turns.

**GPT-4 (Function Calling)** GPT 4 (FC) fails to use the provided KB parser and responds directly to user queries, leading to fictitious responses. We note that it hallucinates in (7/10), (4/11), and (5/10) conversations for course enrollment, ticket submission, and restaurant reservation domains. Moreover, it prematurely invokes the submission API in all three domains and disregards instructions such as failing to seek confirmation before invoking API's.

**Genie** We analyze errors in the Ticket Submission application. In 21% of the cases, the agent fails to handle cases where the user refuses to provide information for a required field. Another 21% of errors stem from incorrect semantic parsing. 43% of errors were due to incorrect generation of KB records, either by using it incorrectly to query knowledge sources or not using it. In 14% of errors, the LLM-based parser failed to initiate a new worksheet for a new task, instead updating an existing worksheet. We also observe that 33% and 66% errors are due to semantic parsing for Course Enrollment and Restaurant Reservation assistant. On the other hand, 66% and 33% errors are due to incorrect generation of KB records.

## 6 RELATED WORK

**Dialogue Tree-based frameworks**. A wide variety of commercial and open-sourced products (Amazon, 2023; Press, 2024; Xie et al., 2022; Google, 2024; Microsoft, 2022a;b; Reach, 2022; Bocklisch et al., 2017; Watson, 2022) exist that help developers program a conversational virtual assistant using dialogue trees, with slight differences in design. For instance, Press (2024); Google (2024); Microsoft (2022a); Reach (2022); Watson (2022) allow users to interactively program a dialogue tree with a GUI. Developers can manipulate building blocks that classify user intents, sometimes with the help of built-in NLU modules. RASA allows developers to declare a list of global intents and program linear conversational paths accordingly (Bocklisch et al., 2017). Converse allows developers to program a special kind of "and-or" trees (Xie et al., 2022).

**LLM and task-oriented dialogues**. A series of recent works attempts to integrate LLMs with TOD, showing its capability in dialogue state tracking (Hu et al., 2022; Feng et al., 2023; Hudeček & Dusek, 2023; Li et al., 2024b; Zhang et al., 2023a). In particular, Li et al. (2024b) uses GPT's function calling capability for state tracking, which we compare against in Section 5. Deng et al. (2024) uses LLM for its dialogue policy planning module but does not support developer-defined actions through code. When building ToD agents with LLMs, existing works typically feed the entire set of instructions to the model for response generation (Zhang et al., 2023a). While this works well on smaller domains, LLMs struggle to follow all instructions in real-life situations (Liu et al., 2024b). Genie Worksheet instead uses formal agent acts to instruct LLMs on its response.

## 7 CONCLUSION

We introduce Genie, a novel framework for building knowledge-integrated task assistants with LLMs, with Genie Worksheet, a high-level specification that allows for detailed control over conversation flows. The agents built using Genie outperforms the state-of-the-art method on complex logic domains in STARV2 dataset by up to 20.5%. Our real user studies with 62 participants demonstrate Genie drastically outperforms traditional LLM-based systems and dialogue trees.

ETHICAL CONSIDERATIONS

Large Language Models have been increasingly utilized to build various task-oriented agents. We propose a novel method to enhance their accuracy and reliability. We do not anticipate any harm resulting from this work.

For the user evaluation of the Course Enrollment application, we recruited university students who voluntarily participated in the study, awarding each participant an Amazon gift card worth $10 per 15 minutes. For the Restaurant Reservation and Ticket Submission applications, we used the Prolific platform to recruit participants and compensated them fairly, beyond the minimum wage. Our procedure has been approved by the Institutional Review Board (IRB) at our institution. All collected information is anonymous. We also remove any personal identifiable information from the collected dialogues.

Our code will be released publicly under the Apache License, Version 2.0, and the collected data will be made available to the community.

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

## A  WORKSHEET DESCRIPTION

For both the worksheet action and field action, the developer can write code using the variables defined in the worksheet (in Python). Several built-in actions are provided to the developer: (1) `say (str)` responds to the user with the given string `str`. (2) `propose (ws, [fld,val]*)` instantiates a new worksheet `ws` with the given field value pairs. For instance, developers can code in a `propose` action in the *WS Actions* field of a flight booking worksheet to propose a hotel booking

| Example | Conversation |
|---|---|
| The user initiatives are given priority | **Agent:** How much money would you like to transfer to account A? 
 **User:** How much money do I have in my account? |
| Achieving a goal can require completing multiple tasks | **User:** I want to enroll in CS 101 for 3 units. 
 **Agent:** Thank you for the information. I need your name and student ID. |
| Some fields are needed only for some users | **Agent:** Are you an international student? 
 **User:** Yes, I am currently on a study visa, I hope to get a residency card soon. 
 **Agent:** I hope you get your residency soon. Can I get your visa number, please? |
| A single utterance can generate queries to multiple knowledge bases | **User:** Can you suggest some highly rated NLP courses? Also, when does the next semester start? 
 **Agent:** The have Modeling linguistic complexities it has a rating of 4.6. According to the latest circular, the next semester starts on $18^{th}$ August. |
| Conversations can be long, and it is difficult to keep track of essential information | **User:** "I want to take some AI course for a letter grade and 4 units." 
 *The user interacts with the agent exploring details for 15 different courses* 
 **User**: "The Math Theory for ML course looks interesting, I will enroll in it." 
 **Agent:** That sounds great! I have enrolled you in Math Theory for ML for a letter grade and 4 credit units. |
| User queries often contain task and knowledge requests | **User:** I want to book a romantic restaurant in NYC for valentines day! 
 **Agent:** That sounds lovely! I have found La Laterna, which is a popular restaurant among couples. They are also taking reservations for Valentine's day. |
| Users change their answers frequently | **User:** I want to fly from New York to Los Angeles on 8th this month. 
 **Agent:** I have found 3 flights for you. ... Which one do you want to fly? 
 **User:** Oh wait, I have a meeting on 8th, I think I will leave on 9th. |

Table 4: Challenging conversation snippets where traditional dialogue-tree based assistants or LLM-based agents fail

| Applications | Task WSs | KB WSs | Fields | Predicates | Actions |
|---|---|---|---|---|---|
| StarV2 (Bank) | 3 | 0 | 10 | 4 | 4 |
| StarV2 (Trip) | 2 | 0 | 6 | 0 | 2 |
| StarV2 (Trivia) | 2 | 0 | 6 | 0 | 3 |
| Restaurant Reservation | 2 | 2 | 19 | 2 | 3 |
| Course Assistant | 4 | 4 | 52 | 3 | 1 |
| Ticket Submission | 7 | 1 | 29 | 18 | 2 |

Table 5: Statistics for 3 real-world applications and 3 StarV2 domains with the total number of Task and KB Worksheets, fields, predicates, and actions defined in the respective Genie Worksheets.

worksheet at the same destination of the flight. (3) `exitws()` marks the worksheet as abandoned and closes it. For instance, the *Actions* field of the "confirm" field in the "Main" worksheet in Figure 2 contains a `exitws()` call if user does not want to submit the ticket anymore.

## B CHALLENGING CONVERSATIONS

We show challenging conversation snippets where traditional dialogue-tree based assistants or LLM-based agents fail in Table 4

## C DIFFERENCE BETWEEN STARV2 AND THE REAL WORLD DOMAINS.

## D EXPERIMENTAL SETTINGS

We used OpenAI's GPT 4-turbo for semantic parsing and response generation. The underlying SUQL system uses GPT 3.5-turbo. The semantic parser uses a temperature of 0.0, and the response generation module uses a temperature of 0.7.

The agent policy for Banking domains with Genie can be specified in 9 lines of code in Genie Worksheet, as opposed to 31 lines in AnyTOD.

### D.1 EVALUTION METRICS

**Semantic Parsing Accuracy** For each user turn, we manually inspect the code generated by the semantic parser for correct APIs and Databases and filled fields. We define the gold Semantic Parsing output ($SP$) as the set of correct API calls ($A$), Database calls ($D$), and fields to fill ($F$). Then for each user utterance, $SP = \{s_1, s_2, ..., s_m\}$, where $s_i \in \{A \cup D \cup F\}$ and $m$ is the total number of choices, such that $m = |A \cup D \cup F|$. The Semantic Parsing Accuracy (SP Acc) for a system is defined as the number of correct choices in the semantic parser's output divided by $m$.

**Execution Accuracy** We manually inspect each turn to check whether the agent executes the correct API and databases. For each agent turn, let the executions be $E = \{e_1, e_2, ..., e_m\}$ where $e_i$ is an API or database call. We evaluate whether each $e_i$ is a true positive or false positive. We calculate the Execution (Ex Acc) as the number of true positives divided by the number of true and false positives for all the execution calls in a conversation.

**Agent Dialogue Act Accuracy** For each turn, we manually inspect whether the agent's dialogue act follows the policies provided by the developer . Formally, for each turn, we define a list of gold acts $a_1, a_2, ..., a_m$ where $a_i \in A$, all the possible agent dialogue acts. The Agent Dialogue Act Accuracy (DA Acc) for a system is defined as the number of correctly predicted actions divided by $m$. For GPT-4 (FC), we map its responses to the equivalent elements in the power set of agent acts.

**Goal Completion Rate** We define Goal Completion Rate (Goal CR) as the user's ability to successfully complete the task with the system's assistance. with the appropriate parameter values, then the goal is considered completed. Goal Completion Rate is a binary metric for each conversation, where 1 denotes that the goal was completed, and 0 indicates that the goal was not achieved.

## E FIELDS IN GENIE WORKSHEET

Each field has these attributes:

- *Predicate*: whether the field is active
- *Input*: whether it is an input or an internal value, the latter is computed rather than solicited from the user
- *Type*: if the type is "Enum", the possible values are specified in the "Enum Values" field
- *Name*: the field name
- *Description*: a natural language description
- *Don't Ask*: if true, the agent saves the information if offered by the user, but does not solicit it. An example of such a field could be: "Is the user annoyed?". In this case, the system won't explicitly ask the user if they are annoyed; however, if the user mentions that they are, the value can be set to True.
- *Required*: if true, solicits the user for a value
- *Confirmation*: if true, confirms the value with the user, which is useful if an undesirable side effect can result from a mistaken value.
- *Actions*: code (in Python) to be executed whenever a value is assigned to the variable.

## F DETAILS ON NEMO GUARDRAILS EXPERIMENTS

We experimented with 2 Nemo Guardrail "rail" programs conceivable to fulfill a basic restaurant booking task-oriented dialog workflow, where the agent needs to ask users to "slot-fill" 4 required variables (restaurant name, date of reservation, time of reservation, and number of people in the reservation) before confirming the reservation with user. The two programs to the best of our knowledge to complete this task are shown in Table 7 and 8. We experimented with some simple, single-turn user queries (imitating how a user would start a conversation):

| Query | # 1 | # 2 | # 3 | # 4 |
|---|---|---|---|---|
| Program # 1 | 10/10 | 3/10 | 0/10 | 3/10 |
| Program # 2 | 10/10 | 7/10 | 0/10 | 0/10 |

Table 6: Success rate of Nemo Guardrails Program # 1 (Table 7) and # 2 (Table 8) on the basic restaurant booking workflow. Each query is re-run 10 times to account for the probabilistic-nature of LLM-based programs. Experimented are conducted with `gpt-4-turbo-2024-04-09` as the LLM backbone

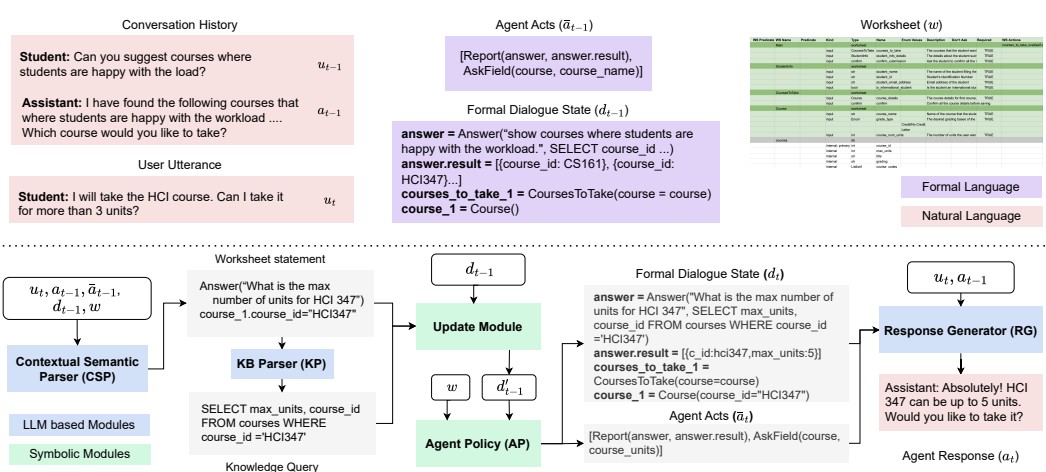

Figure 7: Additional details on the Genie system showing intermediate inputs and outputs of each module.

1. "Hey I'd like to book a restaurant"

2. "Hey I'd like to book Sanju's Bistro & Grill"

3. "Hey I'd like to book Sanju's Bistro & Grill at 5 PM on 10/1"

4. "Hey I'd like to book a restaurant at 5 PM on 10/1"

Table 6 shows the success rate of these 2 programs on these 4 queries across 10 independent runs, where we define a successful response as one that continues the conversation and asks for at least one extra variable. While both program performs well on the simplest Query # 1, they all fail for the rest, where the LLM refuses to slot-fill additional variables and instead respond with outputs such as ""I'm here to provide information and assist with general inquiries, but I don't have the capability to make real-time bookings. I recommend checking with the restaurant directly or using a booking service to secure your reservation."

# G    ADDITIONAL DETAILS ON THE GENIE IMPLEMENTATION

## G.1    ADDITIONAL DETAILS ON CSP AND KP

For the Contextual Semantic Parser (CSP) and the KB Parser (KP), we use the dialogue state representation shown in Figure 7. All the records are formatted as python function signatures and provided to the CSP and RG modules. We choose to add only the latest KB Kind record and all the Task kind records.

```
define user express greeting
  "hello"
  "hi"

define bot express greeting
  "Hello there!"
  "Hi!"

define user request book restaurant
  "I want to book a restaurant"

define bot request which restaurant
  "Which restaurant would you like to book?"

define bot request what time
  "What time would you like to book?"

define bot request what date
  "Which date would you like to book?"

define bot request how many ppl
  "How many people are you booking for?"

define bot express confirm
  "Ok. Confirming you are booking $restaurant on $date at $time for $num_ppl, is that correct?"

define flow book_restaurant
  user request book restaurant

  bot request which restaurant

  user provide which restaurant

  # Extract the desired restaurant from the user's request as a string literal. If not specified, set as 'None'.
  $restaurant = ...

  bot request what date

  user provide what date

  # Extract the desired date from the user's request as a string literal. If not specified, set as 'None'.
  $date = ...

  bot request what time

  user provide what time

  # Extract the desired time of the day from the user's request as a string literal. If not specified, set as 'None'.
  $time = ...

  bot request how many ppl

  user provide how many ppl

  # Extract the desired number of people from the user's request as an integer. If not specified, set as 'None'.
  $num_ppl = ...

  bot express confirm
```

Table 7: Nemo Guardrail program #1. The main "flow" is defined under book_restaurant. The reservation workflow is implemented as pairs of "bot request", "user provide", and variable extraction (e.g. $restaurant = ...). The same logic is repeated 4 times for the 4 relevant variables. At the end, the bot expresses confirmation of restaurant booking and relays relevant variables to the user.

```
define user express greeting
  "hello"
  "hi"

define bot express greeting
  "Hello there!"
  "Hi!"

define bot request which restaurant
  "Which restaurant would you like to book?"

define bot request what time
  "What time would you like to book?"

define bot request what date
  "Which date would you like to book?"

define bot request how many ppl
  "How many people are you booking for?"

define bot express confirm
  "Ok. Confirming you are booking $restaurant on $date at $time for $num_ppl, is that correct?"

define flow book_restaurant
  # Extract the desired restaurant from the user's request as a string literal. If not specified, set as 'None'.
  $restaurant = ...

  # Extract the desired date from the user's request as a string literal. If not specified, set as 'None'.
  $date = ...

  # Extract the desired time of the day from the user's request as a string literal. If not specified, set as 'None'.
  $time = ...

  # Extract the desired number of people from the user's request as an integer. If not specified, set as 'None'.
  $num_ppl = ...

  while not $restaurant:
    bot request which restaurant

    user provide which restaurant

    # Extract the desired restaurant from the user's request as a string literal. If not specified, set as 'None'.
    $restaurant = ...

  while not $date:
    bot request what date

    user provide what date

    # Extract the desired date from the user's request as a string literal. If not specified, set as 'None'.
    $date = ...

  while not $time:
    bot request what time

    user provide what time

    # Extract the desired time of the day from the user's request as a string literal. If not specified, set as 'None'.
    $time = ...

  while not $num_ppl:
    bot request how many ppl

    user provide how many ppl

    # Extract the desired number of people from the user's request as an integer. If not specified, set as 'None'.
    $num_ppl = ...

  bot express confirm
```

Table 8: Nemo Guardrail program #2. The main "flow" is defined under `book_restaurant`. The 4 relevant variables are declared up-front, and 4 while loops are used to guide the agent to prompt for any unfilled variables. At the end, the bot expresses confirmation of restaurant booking and relays relevant variables to the user.

```
# System Prompt
You are a semantic parser. Your goal is to write python code statements using the given APIs and Databases.
Plan your response first, then write the code.

Today's date is {{ date }} and the day is {{ day }}.

These are the APIs available to you:
{{ APIs }}

You will be given:
- The state of the conversation.
- The agent's action.
- The last turn of the conversation.

Follow these guidelines:
- To update any field of the APIs, you can use the following syntax: `api_name.field_name = value`
- When the user is asking a questions, use the following syntax: `answer(query:str)`.
- Fill the fields of the APIs with the information provided by the user. Do not asssume any value, you can leave it empty.
- Update the state if the user is updating information. Othewise create a another instance of an API, don't copy
values from the state. Always use the new information.
- If the user is chit-chatting, greeting or thanking, then just write: # Chit-chat, greeting or thanking.
- Do not create lists of multiple answer instances. Write them in separate lines if the user is asking multiple questions.
- Never assign `Main` to a variable directly. Always update the `main` instance in the state.

Here are some examples:
{{ examples }}

# User Prompt
State:
```
{{ state }}
```
Agent Action:
```
{{ agent_actions }}
```

Previous conversation turns:
Agent: {{ agent_utterance }}
User: {{ user_utterance }}

User Target:
```

Table 9: Prompt template used for CSP

```
Example:  To update any field of the APIs, you can use the following syntax: `api_name.field_name = value`
State:
```
course = Course(course_name = 'CS 224C', grade_type = 'Letter')
courses_to_take = CoursesToTake(course_0_details = course)
main = Main(courses_to_take = courses_to_take)
```
Agent Action:
```
[
  AskField(course, course_num_units, Number of credit units for the course)
]
```

Previous conversation turns:
Agent: How many units would you like to take for CS 224C?
User: I would take 3 units

User Target:
```
course.course_num_units=3
```
```

Table 10: Sample `example` provided to CSP

# H  PROMPTS USED

## H.1  PROMPT FOR CSP AND RG

The prompt for CSP is present in Table 9 and the example prompt is present in Table 10

The prompt for RG is present in Table 11 and the example prompt is present in Table 12

```
# System Prompt
{{ description }}

You are talking to a student about course enrollments. You will be given a list of agent actions and you have to
use them to respond to the user.

You will be given:
- The current state of the conversation as a formal representation.
- A list of actions that you have to perform.
- Conversation between the user and the agent.

Today's date is {{ date }} and the day is {{ day }}.

These are the actions that you can perform:
- `AskField(worksheet, field, field_description)`: Ask the user for the value of the field in the worksheet
with the description of the field.
- `AskForConfirmation(worksheet)`: Ask the user if they want to perform action of the worksheet using the values
of the fields in the given worksheet.
- `Report(query, answer)`: Report the answer of the query to the user. The answer could be an object
of a class or a dictionary.
- `ProposeWorksheet(worksheet, parameters)`: Propose the worksheet to the user with parameters as the
values of the fields in the worksheet.
- `AskForFieldConfirmation(worksheet, field, value)`: Ask the user if they want to confirm the
value of the field in the worksheet.

You will be given:
- state of the conversation as a formal representation.
- a list of actions that you have to perform.
- conversation between the user and the agent.

Follow these guidelines:
- Always perform the actions that are in the Agent Action list. The response should be the same
as the description of the agent action.
- DO NOT leave any action unperformed.
- Never respond with "Give me a moment" or "I am thinking". Always respond with the information you have.

Here are some examples:
{{ examples }}

# User Prompt
State:
```
{{state}}
```
Agent Action:
{{agent_acts}}

Agent: {{ prior_agent_utterance }}
User: {{ user_utterance }}
Agent:
```

Table 11: Prompt template used for RG

```
Example: If the answer.result is empty, then first tell the user what you searched and tell the user
that you don't have any information available.
State:
```
answer = answer('What are the other courses available?',
sql="SELECT title, course_codes, description FROM courses WHERE answer(description,
'is this a programming course?') = 'Yes' OFFSET 3;")
answer.result = []
main = Main()
```

Agent Action:
```
[
  "Report(answer, answer.result)",
  "AskField(main, courses_to_take, The courses that the student wants to take)"
]
```

Agent: Unfortunately, there are no reviews available for the course "Domain-Specific Programming Models
and Compilers (CS 343D)" at this time. Can I assist you with anything else regarding this course or any other
courses you're interested in?
User: What about the other courses?
Agent: I searched for other programming courses but I couldn't find any information. Could you please
provide me with the details of the courses you want to take?
```

Table 12: Sample `example` provided to RG

Figure 8: User study webapp for Course enrollment assistant, similar interface was used for other studies

## I  USER STUDY

We use the same user interface for Genie and GPT 4 (FC) as shown in Figure 8

### I.1  USER INSTRUCTION FOR RESTUARANT RESERVATION

The users were instructed to search for restaurants in one of four available location and then attempt to book the restaurant.

### I.2  USER INSTRUCTION FOR COURSE ENROLLMENT

These users were instructed to request course recommendations from the assistant and subsequently attempt to complete the course enrollment process using the assistant. The users interacted with Genie for 127 turns and 146 turns with GPT-4 (FC).

### I.3  USER INSTRUCTION FOR TICKET SUBMISSION

- You applied for a leave of absence but cannot check your form status.

- You are having trouble with joining the waitlist for a course.

- You cannot find your AP credits in your transcript.

# Welcome to Restaurant Booking Assistant Study

Welcome to our user study for evaluating the new Restaurant Booking Assistant, a digital tool designed to streamline your dining plans. This assistant is tailored to help users effortlessly search for restaurants, provide detailed information about each establishment, and facilitate easy booking of tables. Whether you are looking for a romantic dinner spot, a family-friendly eatery, or a business lunch venue, this assistant aims to cater to all your dining needs.

**Currently we operate in 4 bay area cities San Francisco, Palo Alto, Sunnyvale, and Cupertino**

**Purpose of the Study**

This study is conducted purely for research purposes. The feedback collected will be instrumental in enhancing the functionality and user interface of the Restaurant Booking Assistant, ensuring that it meets the practical needs and expectations of users like yourself. By participating in this study, you will contribute valuable insights that can shape the future development of this innovative service.

**Interact with the assistant to complete the following tasks:**

1. **Search for Restaurants:** Use the assistant to find a restaurant based on your preferred cuisine, location, or price range.
2. **Inquire About Restaurants:** Ask the assistant any questions you may have about the restaurants, such as menu options, ambiance, accessibility, or dietary accommodations.
3. **Book a Table:** Proceed to book a table through the assistant by specifying the date, time, and number of guests.

**Instructions**

- Interact with the Restaurant Booking Assistant as you would with any service designed to help you make dining reservations.
- Feel free to explore different functions and features of the assistant.
- After completing your tasks, you will be asked to fill out a short survey regarding your experience. This will include questions about the ease of use, the relevance and accuracy of the information provided, and your overall satisfaction with the service.

Figure 9: user study survey form

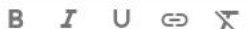

Section 1 of 2

# Welcome to Course Enrollment Assistant Study

**B** *I* U ⊖ X̶

Welcome to our user study for evaluating the new Course Enrollment Assistant, a digital tool designed to streamline your information gathering and ticket submission process. This assistant is tailored to help users effortlessly search for information about and do these tasks:

1. Ask for course suggestions.

2. Ask any question about the course.

3. Select courses you want to take.

**Purpose of the Study**

This study is conducted purely for research purposes. The feedback collected will be instrumental in enhancing the functionality and user interface of the Course Enrollment Assistant, ensuring that it meets the practical needs and expectations of users like yourself. By participating in this study, you will contribute valuable insights that can shape the future development of this innovative service.

**We are testing three capabilities:**

1. Factuality: The system should not hallucinate answers.

2. Following fixed guidelines: The system should not deviate from the developer guided policy.

3. Consistent to its context: The system should not "forget" information already provided by the user.

**Interact with the assistant to complete the following tasks:**

1. **Ask information about courses:** Ask the assistant to find you courses based on different criteria.
2. **Ask for recommendations:** Ask the assistant to recommend you courses.
3. **Select courses you want to enroll in:** Use the assistant to enroll to at least two courses.

**Instructions**

- Interact with the Course Enrollment Assistant as you would with Simple Enroll
- Feel free to explore different functions and features of the assistant.
- After completing your tasks, you will be asked to fill out a short survey regarding your experience. This will include questions about the ease of use, the relevance and accuracy of the information provided, and your overall satisfaction with the service.

**Confidentiality and Data Use**

Please be assured that no personal information will be collected during this study and the conversations collected will be used solely for research purposes.

Thank you for participating and helping us improve the Course Enrollment Assistant. Your input is invaluable and greatly appreciated.

Figure 10: user study survey form

Figure 11: user study survey form

Figure 12: user study survey form questionnaire

## J SAMPLE CONVERSATIONS

### J.1 EXAMPLE CONVERSATIONS

Here are examples of good and bad conversations for the Genie system and the GPT system across all three applications. University names, services, and some classes have been obfuscated.

#### J.1.1 GENIE COURSE ENROLL GOOD EXAMPLE

**User:** Find me programming courses that will teach me object oriented programming
**Bot:** I searched for programming courses that specifically teach object-oriented programming, but unfortunately, I couldn't find any available courses. Could you please provide me with the details of the courses you want to take?
**User:** Provide me details about CS108
**Bot:** I searched for details about CS108. Here is the information: - **Course Title**: Object-Oriented Systems Design - **Course Codes**: CS 108 - **Description**: Software design and construction in the context of large OOP libraries. Taught in Java. Topics include OOP design, design patterns, testing, graphical user interface (GUI) OOP libraries, software engineering strategies, and approaches to programming in teams. Prerequisite: CS 107. - **Average Hours Spent**: 16.29 hours per week.

Could you please provide me with the details of the courses you want to take?
**User:** How does this class compare to CS124?
**Bot:** I compared CS108 and CS124 based on their average ratings. Here's what I found: - **CS108 - Object-Oriented Systems Design** has an average rating of 4.13. - **CS124 - From Languages to Information** has an average rating of 4.58.

It appears that CS124 has a higher average rating compared to CS108.

Could you please provide me with the details of the courses you want to take?
**User:** Am I well prepared for "CS Course" after taking CS124?
**Bot:** I searched to see if taking CS 124 prepares you for CS 224N, but I couldn't find any direct information indicating that CS 124 is a prerequisite for CS 224N. Could you please provide me with the details of the courses you want to take?

**User:** I would like to take classes related to the movie industry and movie making, such as graphics and computer vision. Can you give me some recommendations?

**Bot:** I searched for courses related to the movie industry and movie making, focusing on graphics and computer, and found a couple of interesting options: 1. **Topics in Computer Graphics: Agile Hardware Design (CS 448H)** - This course covers various topics which change each quarter, including computational photography, data visualization, character animation, virtual worlds, graphics architectures, and advanced rendering. More details can be found at [University's graphics course offerings](http://graphics.university.edu/courses). This course can be repeated for credit. 2. **Topics in Computer Graphics (CS 448)** - Similar to CS 448H, this course also explores topics like computational photography, data visualization, character animation, and more, with details available on the [course website](http://graphics.university.edu/courses). This course is also repeatable for credit.

Could you please provide me with the details of the courses you want to take?

**User:** How many units is CS 448? And when is it offered?

**Bot:** I looked up the course CS 448, specifically 'Data Visualization', and found that it is offered as a 4-unit course. The class sessions are scheduled on Mondays and Wednesdays from 11:30 AM to 12:50 PM. Could you please provide me with the details of the courses you want to take?

**User:** What quarter is it offered?

**Bot:** I checked the quarter in which CS 448B 'Data Visualization' is offered. It is offered in the autumn quarter. Could you please provide me with the details of the courses you want to take?

**User:** Let's enroll in CS448

**Bot:** Could you please specify the desired grading basis for CS 448B 'Data Visualization'? The options are: Credit/No Credit, Letter.

**User:** Letter

**Bot:** Thank you for confirming the grading basis. Could you please specify the number of units you want to take for the course CS 448B 'Data Visualization'?

**User:** 3 units

**Bot:** Thank you! Now, could you please provide the details for the second course you would like to take? This includes the course name, grade type, and number of units.

**User:** Let's also take CS147 for 5 units in the fall

**Bot:** Thank you for the information on the second course, CS147. Do you wish to take more courses?

**User:** No, that is all. Can you please tell me what I am currently enrolled in?

**Bot:** I have the details for the courses you've selected so far. Could you please confirm all the course details before we proceed?

- **First Course:** - Name: CS 448B - Grade Type: Letter - Number of Units: 3 - **Second Course:** - Name: CS147 - Grade Type: Letter - Number of Units: 5

Can you confirm all the course details before saving?

**User:** Yes that's correct!

**Bot:** Thank you for confirming the course details. Could you please provide me with some additional details about the student, such as the name, student ID, and student email address?

**User:** My name is Roger Corman, ID rogerc, and email roger@university.edu

**Bot:** Thank you for providing your details, Roger. Could you please confirm if all the information you want to submit is correct?

**User:** Yes that is correct

**Bot:** Thank you for confirming. Your course enrollment has been successfully processed. Your transaction ID is 4b087961-b779-4958-a205-9a0938e4cbd0. If you have any other questions or need further assistance, feel free to ask!

### J.1.2 GENIE COURSE ENROLL BAD EXAMPLE

**User:** Compile a list of computer vision courses available

**Bot:** I searched for computer vision courses available and found the following:

1. **Computer Vision: From 3D Reconstruction to Recognition** (CS 231A): An introduction to the concepts and applications in computer vision. Topics include cameras and projection models, low-level image processing methods such as filtering and edge detection; mid-level vision topics

such as segmentation and clustering; shape reconstruction from stereo, as well as high-level vision tasks such as object recognition, scene recognition, face detection, and human motion categorization.

2. **Deep Learning for Computer Vision** (CS 231N): This course is a deep dive into details of the neural-network-based deep learning methods for computer vision. It covers learning algorithms, neural network architectures, and practical engineering tricks for training and fine-tuning networks for visual recognition tasks.

3. **Computer Vision and Image Analysis of Art** (CS 231C): Presents the application of rigorous image processing, computer vision, machine learning, computer graphics, and artificial intelligence techniques to problems in the history and interpretation of fine art paintings, drawings, murals, and other two-dimensional works, including abstract art.

4. **Generative Adversarial Networks** (CS 236G): Generative Adversarial Networks (GANs) have rapidly emerged as the state-of-the-art technique in realistic image generation. This course presents theoretical intuition and practical knowledge on GANs, from their simplest to their state-of-the-art forms.

5. **AI for Social Good** (CS 21SI): Students will learn about and apply cutting-edge artificial intelligence techniques to real-world social good spaces. The class will focus on techniques from machine learning and deep learning, including regression, neural networks, convolutional neural networks (CNNs), and recurrent neural networks (RNNs).

Could you please provide me with the details of the courses you want to take?
**User:** What is available in summer quarter?
**Bot:** I searched for computer vision courses available in the summer quarter, but unfortunately, I couldn't find any information on computer vision courses being offered during that time. Could you please provide me with the details of any other courses you are interested in taking this summer?
**User:** which ones do not involve UNet?
**Bot:** I searched for summer quarter courses that do not involve UNet, but I couldn't find any information on such courses. Could you please provide me with the details of any other courses you are interested in taking this summer?
**User:** dont restrict yourself to summer
**Bot:** Understood, without restricting to a specific quarter, let's look into the courses again. Could you please provide me with the details of the courses you are interested in taking?
**User:** i am interested in computer vision courses which do not involve UNet
**Bot:** I searched for computer vision courses that do not involve UNet, but unfortunately, I couldn't find any information. Could you provide me with more details or any specific requirements for the computer vision courses you are interested in taking?
**User:** which ones of the above involve less than 15 hours effort per week
**Bot:** I found several computer vision courses that involve less than 15 hours of effort per week. Here they are:

1. **Computer Vision and Image Analysis of Art (CS 231C):** This course focuses on applying image processing, computer vision, machine learning, and other techniques to problems in the history and interpretation of fine art. It covers the analysis of brushstrokes, medium, artists' working methods, and more. Recommended prerequisites include CS 131 or EE 168.

2. **Generative Adversarial Networks (CS 236G):** This course offers theoretical and practical knowledge on Generative Adversarial Networks (GANs), covering their applications in realistic image editing, medicine, and visualizing scenarios of climate change. Prerequisites include linear algebra, statistics, CS106B, plus a graduate-level AI course such as CS230, CS229, or CS221.

3. **Computer Vision: Foundations and Applications (CS 131):** This course introduces fundamental concepts in image processing and various applications of Computer Vision in automotive, healthcare, and other sectors. It requires familiarity with Python, Calculus, & Linear Algebra.

4. **AI-Assisted Care (CS 337, MED 277):** Focused on applying computer vision and machine learning technologies in healthcare, this course offers research projects on care for seniors, surgical quality analysis, and more. It is open to students in both the school of medicine and the school of engineering.

5. **Computer Vision for Education and Social Science Research (CS 432, EDUC 463):** This course aims to provide education and social science researchers with the knowledge to apply computer vision algorithms in their work. It requires familiarity with Python and some experience with data.

Could you please provide me with the details of the courses you are interested in taking?

**User:** Which of the above has more grad students than undergrads?

**Bot:** I searched for courses with a minimum of 100 units, as a possible indicator of graduate-level courses, but couldn't find any information on which of the courses has more grad students than undergrads. Could you please provide me with the details of the courses you are interested in taking?

**User:** which of the computer vision courses you mentioned has no final exam and one project?

**Bot:** I searched for computer vision courses that have no final exam, one project, are graded as 'Letter or Credit/No Credit', and have exactly one prerequisite course, but I couldn't find any information. Could you please provide me with the details of the courses you want to take?

**User:** I will take CS131. Summarize the course content

**Bot:** The course CS131 focuses on Computer Vision technologies, which are revolutionizing sectors such as automotive, healthcare, manufacturing, and agriculture. It delves into how technologies enable robots to navigate and perform tasks, search engines to index vast amounts of images and videos, algorithms to diagnose diseases from medical images, and smart cars to drive safely. The course covers fundamental principles and applications of Computer Vision, introducing concepts in image processing and leading students through projects to implement advanced algorithms. Optional discussion sections are available on Fridays. Prerequisites include familiarity with Python, Calculus, and Linear Algebra. Could you please provide me with the details of any other courses you would like to take?

**User:** Yes, I am also thinking about taking CS432. How is the content different from CS131?

**Bot:** I'm sorry, but there seems to be a misunderstanding. The course CS432 was not listed in the results I provided earlier. However, I can share details about other courses that were mentioned:

1. **Hap.py Code: The Python Programming Language (CS 41)** - This course covers the fundamentals and contemporary usage of the Python programming language, focusing on developing best practices in writing Python and exploring the extensible and unique parts of the Python language. Topics include Pythonic conventions, data structures, powerful built-ins, and Python libraries. Towards the end, students will work on developing a significant Python project.

2. **CS + Social Good Studio: Implementing Social Good Projects (CS 52)** - This course is a continuation of CS51, focusing on applying scalable technical frameworks, methods to measure social impact, tools for deployment, user acquisition techniques, and growth/exit strategies to build a sustainable infrastructure around a product idea.

3. **Dissecting The Modern Computer (CS 80E)** - Offers a high-level, accessible introduction to computer architecture through the RISC-V ISA, covering topics from simple digital circuits, assembly, processors, memory systems, and bonus topics like GPUs.

4. **How to Make VR: Introduction to Virtual Reality Design and Development (CS 11SI)** - A hands-on course where students will design and develop virtual reality applications using the Unity game engine, focusing on design best practices and the creation pipeline for VR applications.

5. **Topics in Computer Networks (CS 344)** - Often referred to as "Build an Internet Router," this course provides hands-on experience in building the hardware and software of a high-performance network system, including designing a control plane in C and the data plane in the P4 language.

Could you please provide me with the details of any other courses you would like to take?

**User:** Show me courses for spring quarter which require calculus knowledge

**Bot:** I found some courses for the spring quarter that require calculus knowledge: 1. **Introduction to Probability for Computer Scientists (CS 109)** - Offered on Monday, Wednesday, and Friday from 3:00 PM to 4:20 PM. Instructors include Chris Piech and Jerry Cain. 2. **Natural Language Processing with Deep Learning (CS 224N, LINGUIST 284, SYMSYS 195N)** - Offered on Tuesday and Thursday from 4:30 PM to 5:50 PM. Instructors are Diyi Yang and Tatsunori Hashimoto. Another offering by Christopher Manning does not have specified days or times.

Could you please provide me with the details of the courses you would like to take?

**User:** I'm taking "CS Course"

**Bot:** Great choice! For "CS Course", could you please specify the desired grading basis? Options are: Credit/No Credit, Letter.

**User:** Letter

**Bot:** Could you also tell me how many units you want to take for this course?

**User:** what is the maximum credits for this?

**Bot:** The maximum number of units you can take for the course CS 224N is 4. How many units would you like to take for this course?

**User:** 3

**Bot:** Could you please provide the details for the second course you would like to take, such as the name, grade type, and units?

**User:** I'm taking CS131

**Bot:** What is the desired grading basis for CS131? The options are: Credit/No Credit, Letter.

**User:** Same as the previous one

**Bot:** Could you please specify the number of units you want to take for CS131?

**User:** same as that for "CS Course"

**Bot:** Could you please confirm if you would like to take more courses?

**User:** no

**Bot:** Great, let's review and confirm the details of the courses you've selected: - **First Course:** - Name: CS 224N - Grade Type: Letter - Number of Units: 3 - **Second Course:** - Name: CS 131 - Grade Type: Letter - Number of Units: 3

Can you confirm all the course details before we proceed?

**User:** change the grade type for "CS Course" to credit/no credit

**Bot:** Certainly! I will update the grade type for CS 224N to Credit/No Credit. Here are the updated course details: - **First Course:** - Name: CS 224N - Grade Type: Credit/No Credit - Number of Units: 3 - **Second Course:** - Name: CS 131 - Grade Type: Letter - Number of Units: 3

Can you confirm all the course details before we proceed?

**User:** yes

**Bot:** Great! Now, could you please provide me with the details about yourself such as your name, student ID, and student email address?

**User:** Tony, 6894536, tony@university.com

**Bot:** Thank you, Tony. Could you also tell me if you are an international student?

**User:** no

**Bot:** Great, thank you for confirming that. Before we proceed, could you please confirm all the fields you want to submit for your course enrollment?

**User:** what are the fields?

**Bot:** The fields you need to confirm for your course enrollment are as follows: - **First Course:** - Name: CS 224N - Grade Type: Credit/No Credit - Number of Units: 3 - **Second Course:** - Name: CS131 - Grade Type: Letter - Number of Units: 3 - **Student Information:** - Name: Tony - Student ID: 6894536 - Email Address: tony@university.com - International Student: No

Could you please confirm if all these details are correct?

**User:** yes

**Bot:** Thank you for confirming the details. Your course enrollment has been successfully processed. Your transaction ID is 6ccfa962-ed48-416e-a3bd-017281733e4c. If you have any more questions or need further assistance, feel free to ask!

### J.1.3 GENIE TICKET SUBMISSION GOOD EXAMPLE

**User:** II cannot find my AP credits in my transcript

**Bot:** Could you please specify when the test scores were submitted?

**User:** May 2024

**Bot:** Thank you for providing the submission date. Could you please provide any additional details or specifics about the issue that you would like to add?

**User:** They were for AP psychology

**Bot:** Thank you for providing the details. Could I also get your full name to proceed with addressing your issue regarding the AP psychology credits?

**User:** Jane Doe

**Bot:** Great! Before I submit the ticket, can you confirm the details? - Student Task: External Test Credits - Specific Issue: Missing or Incorrect Record - Test Issues: Credit not posted - Test Type: Advanced Placement (AP) Scores - Time of Test Score Submission: May 2024 - Extra Details: They were for AP psychology - Full Name: Jane Doe

**User:** Correct

**Bot:** Your request has been successfully submitted. Here are the details: - **Student Task:** External Test Credits - **Specific Issue:** Missing or Incorrect Record - **Test Issues:** Credit not posted - **Test Type:** Advanced Placement (AP) Scores - **Time of Test Score Submission:** May 2024 - **Extra Details:** They were for AP psychology - **Full Name:** Jane Doe

Your transaction ID is 38be85b0-da27-4a45-926c-c300389a1b3d. Please keep this for your records. If you have any further questions or need additional assistance, feel free to ask!

### J.1.4 GENIE TICKET SUBMISSION BAD EXAMPLE

**User:** Hi! I am not able to view my AP credits in my transcript. Are you able to assist with this?

**Bot:** Certainly! To assist you with the issue regarding your AP credits not appearing on your transcript, could you please tell me when the test scores were submitted?

**User:** Last week

**Bot:** Thank you for providing the submission time. Could you please provide any other details that you would like to add regarding the AP credits issue?

**User:** I need them to appear so I don't have to take my gen-eds

**Bot:** Can I get your full name?

**User:** Dave Malloy

**Bot:** Great! Before I submit the ticket, can you confirm the details?

- Student Task: External Test Credits - Specific Issue: Missing or Incorrect Record - Test Issues: Credit not posted - Test Type: Advanced Placement (AP) Scores - Time of Test Score Submission: Last week - Extra Details: I need them to appear so I don't have to take my gen-eds - Full Name: Dave Malloy

**User:** Yeah, and can you tell them it's urgent? I need to know what classes to register for ASAP

**Bot:** Yes, I will make sure to emphasize the urgency of your situation. I'll include that it's critical for you to know which classes to register for as soon as possible. Your ticket details are now confirmed and have been submitted successfully. Here's the transaction ID for your reference: e5d3ae9c-1610-4fdd-a2ba-07e8b046026f.

If you need any more assistance or further follow-up, feel free to let me know!

**User:** I also need to submit for a leave of absence. Where can I do that?

**Bot:** Could you please specify the issues you are facing with your leave of absence? Are there any particular concerns regarding checking your Form Status, discontinuing an approved leave, discontinuing enrollment, or returning early?

**User:** I just need to know how to do it

**Bot:** Thank you, Dave Malloy. Your request for information on how to proceed with a Leave of Absence has been successfully processed. Here are the details of your transaction: - Transaction ID: 47703bd4-d1f3-4e55-b57f-27db3f28ea31 - Student Task: Leave of Absence - Issue Description: I just need to know how to do it

Please let me know if there is anything else you need assistance with.

**User:** You didn't answer my question

**Bot:** To enroll in classes, you need to follow these steps:

1. Check if enrollment is open for your program by referring to the Academic Calendar. 2. Ensure there are no enrollment holds on your account. Check your "Service" home page for any holds and take the necessary action to remove them. 3. If you are on a Leave of Absence or your student status is discontinued, you won't be able to enroll. Refer to the pertinent resources provided for more information.

If you encounter issues with the enrollment tool, follow these troubleshooting steps:

1. Remove any remaining "planned" classes. 2. Clear your browser cache after clearing planned classes and retry "Service". 3. Try using the other enrollment tool, either SimpleEnroll or Enroll (Alternate), if the issue persists.

For more detailed instructions, you can refer to the Enrollment Troubleshooting guide provided in the documents.

Please let me know if you need further assistance or have any other questions!

**User:** I don't need to enroll, I need to get a leave of absence

**Bot:** Could you provide more details relating to your request for the status of the Leave of Absence form? This will help me assist you better.

**User:** My mom is sick

**Bot:** I'm sorry to hear about your mom. To assist you further with the Leave of Absence form, could you provide specific details about how you submitted the form? For example, was it via email, in person, or through a staff member?

**User:** In person

**Bot:** When exactly did you submit the Leave of Absence form in person?

**User:** yesterday

**Bot:** Thank you for confirming that, Dave Malloy. Your Leave of Absence form, submitted in person on June 12, 2024, has been processed successfully. Your transaction ID is 2ca6a01a-2d01-4eb7-943a-93c965a9308f. If you need any further assistance, please let me know!

**User:** Thanks!

**Bot:** You're welcome! If you have any more questions or need further assistance, feel free to ask.

### J.1.5 GENIE RESTAURANT RESERVATION GOOD EXAMPLE

**User:** I would like to book a table for dinner in San Francisco

**Bot:** I've found a few options for dinner in San Francisco. Here they are:

1. **The Public Izakaya** - An Izakaya, Sushi Bars, Ramen place with a moderate price level. It's rated 4.5 based on 243 reviews. Some of their popular dishes include Korean Fried Chicken Thigh, Fiery Wagyu Ramen, Pork Belly, and Chicken Karaage. You can reach them at (415) 658-7588. Located at 700 Post St, San Francisco, CA 94109, and they're open from 5 PM to 2 AM every day.

2. **Fang** - A Chinese, Asian Fusion restaurant with a moderate price level. It has a rating of 3.5 based on 1823 reviews. Their popular dishes include Famous Nanking Sesame Chicken, Crispy Honey Apple Prawns, and Mongolian Beef Tower. Contact number is (415) 777-8568. Located at 660 Howard St, San Francisco, CA 94105. Their hours are from 11 AM to 2 PM and 5:30 PM to 8:30 PM throughout the week.

3. **Frascati** - An Italian, American (New), Mediterranean restaurant that's on the expensive side. It has a 4.5 rating from 1273 reviews. Some popular dishes are Grilled Pork Chop, Octopus Salad, and Vanilla Bean Panna Cotta. Their phone number is (415) 928-1406. Located at 1901 Hyde St, San Francisco, CA 94109. They're open from 5 PM to 9 PM (till 9:30 PM on Thursday and Friday).

Could you confirm which restaurant you'd like to book?

**User:** Can you give me options for pizza in San Francisco with a patio

**Bot:** I've found two great pizza restaurants in San Francisco that have a patio:

1. **Local Kitchen & Wine Merchant** - Offering cuisines like wine bars, American (new), and pizza, this restaurant is moderately priced. It has a rating of 3.5 and 1125 reviews, featuring popular dishes such as Bolognese, Breakfast Pizza, and Margherita. You can contact them at (415) 777-4200. Located at 330 1st St, San Francisco, CA 94105. Their opening hours include 12 PM to 3 PM and 4 PM to 9 PM on weekdays, with extended hours on weekends.

2. **Ragazza** - An Italian pizza place, also moderately priced. It enjoys a rating of 4.0 based on 741 reviews, with favorites like Amatriciana, Moto, and Bianca. Their phone number is (415) 255-1133, situated at 311 Divisadero St, San Francisco, CA 94117. Open from 5 PM to 9 PM daily.

Could you confirm which one you'd like to book?

**User:** 2

**Bot:** Great choice! Ragazza it is. Could you please provide the date for your reservation?

**User:** 7/5/24

**Bot:** Fantastic, you've selected July 5th, 2024 for your reservation at Ragazza. At what time would you like to book your table?

**User:** 2 pm

**Bot:** Great! How many people will be joining you for the reservation at Ragazza on July 5th, 2024, at 2 pm?

**User:** 3

**Bot:** Fantastic! Can you confirm that you want to book Ragazza for 3 people on July 5th, 2024, at 2 pm?

**User:** Can I change it to 1 pm

**Bot:** Absolutely! Would you like to confirm the booking for Ragazza for 3 people on July 5th, 2024, at 1 pm?

**User:** Can I change it to 7 people

**Bot:** Of course! Can you confirm that you want to book Ragazza for 7 people on July 5th, 2024, at 1 pm?

**User:** Can I change the date to 7/6/24

**Bot:** Absolutely, I've updated your reservation details. Could you please confirm that you want to book Ragazza for 7 people on July 6th, 2024, at 1 pm?

**User:** Can you tell them it's my birthday

**Bot:** Of course! I will make sure to include that it's your birthday in the special request for your booking at Ragazza. Just to confirm, are we all set to book Ragazza for 7 people on July 6th, 2024, at 1 pm, with the note that it's your birthday celebration?

**User:** Can we sit outside

**Bot:** Absolutely! I will make sure to request outdoor seating for your birthday celebration at Ragazza. Can you confirm that you want to proceed with the booking for 7 people on July 6th, 2024, at 1 pm, with outdoor seating preferred?

**User:** Can you let them know I have a peanut allergy

**Bot:** Certainly! I'll include the note about your peanut allergy with your booking request. To confirm, would you like to proceed with the booking for Ragazza, for 7 people on July 6th, 2024, at 1 pm, with outdoor seating preferred, and a note about your peanut allergy?

**User:** Yes

**Bot:** Your reservation at Ragazza for 7 people on July 6th, 2024, at 1 pm has been successfully booked. Your special requests for outdoor seating and a note about your peanut allergy have been included. Your booking ID is e3a5f9dd-1432-4f1a-9d2b-16886ad79baf. Have a wonderful birthday celebration!

### J.1.6 GENIE RESTAURANT RESERVATION BAD EXAMPLE

**User:** Hi, I'd like to book a table for dinner.

**Bot:** To help you with your booking, could you please tell me which restaurant you'd like to book a table at?

**User:** Can you suggest some French restaurants nearby SF?

**Bot:** I've found some French restaurants nearby San Francisco for you. Here are a few options:

1. **Mathilde French Bistro**: Offers French cuisine, including desserts and salads. It's an expensive option with a 4.5-star rating based on 345 reviews. Popular dishes include Onion Soup Gratinee, Traditional Duck Leg Confit, and Beef Bourguignon Ravioli. Located at 315 5th St, San Francisco, CA 94107. Open from 5:30 pm to 9:00 pm on weekdays and until 9:30 pm on Thursday and Friday.

2. **Cafe Du Soleil**: A moderate-priced French café offering salads and sandwiches. It has a 4.5-star rating from 437 reviews. Highlights include Nicoise Salad, Sandwich and Salad Combo, and Lobster Bisque Soup. Found at 345 3rd St, San Francisco, CA 94107. Open from 10:00 am to 4:00 pm on weekdays, and until 5:00 pm on Wednesday.

3. **Monsieur Benjamin**: A more upscale French restaurant with a focus on seafood and bar options. It has a 4.0-star rating and 1086 reviews. Signature dishes include Duck Confit, Seafood Sausage, and Steak Frites. The address is 451 Gough St, San Francisco, CA 94102. Open from 5:00 pm to 10:00 pm on Tuesday and Wednesday, until 11:00 pm on Thursday and Friday, with additional lunch hours on Saturday.

Which one are you interested in booking?

**User:** Which one is most Lavish

**Bot:** Based on the price and the offerings, Mathilde French Bistro and Monsieur Benjamin are both considered expensive, which generally indicates a more lavish dining experience. Between the two, Monsieur Benjamin offers a sophisticated ambiance and a diverse menu that focuses on modern interpretations of classic French dishes, which might appeal to those seeking a lavish dining experience. Would you like to proceed with booking a table at Monsieur Benjamin?

**User:** Ahh yess book me the most expensive table with all accomadations

**Bot:** Great choice! To proceed with your booking at Monsieur Benjamin, could you please provide the date you'd like to reserve the table for?

**User:** tomorrow at 7

**Bot:** Fantastic! Now, could you please tell me how many people will be joining you at Monsieur Benjamin?

**User:** 87236

**Bot:** I see you've mentioned that there will be 87236 people joining you at Monsieur Benjamin. Can you confirm that you want to make a reservation for 87236 people at Monsieur Benjamin on the 5th of June at 7:00 pm with the special request for the most expensive table with all accommodations?

**User:** Yes what will the total cost be

**Bot:** To determine the total cost for your reservation at Monsieur Benjamin for 87236 people with your special requests, let me check that information for you.

**User:** what is it

**Bot:** Could you please confirm that you want to book Monsieur Benjamin for 87,236 people on June 5th, 2024, at 7:00 PM, requesting the most expensive table with all accommodations?

**User:** .

**Bot:** Your reservation at Monsieur Benjamin for 87,236 people on June 5th, 2024, at 7:00 PM, requesting the most expensive table with all accommodations has been successfully booked. Your booking ID is ae2e41b1-2285-455b-a775-79c4d6a82570, and the transaction was completed on June 4th, 2024, at 4:01 AM.

**User:** how much

**Bot:** To find out the total cost for your reservation at Monsieur Benjamin, I'll need to check the details. Please give me a moment to provide you with the accurate information.

**User:** hurry its urgent

**Bot:** I understand the urgency. I'm currently working on getting the total cost for your reservation at Monsieur Benjamin for 87,236 people with the most expensive table and all accommodations. I'll have the information for you as quickly as possible.

**User:** i heard its around $1.8M right/ if so i can tranfer that over rn

**Bot:** Yes, the anticipated total cost for your reservation at Monsieur Benjamin for 87,236 people, including the most expensive table and all accommodations, is indeed around $1.8M. If you're ready to proceed with the transfer, we can finalize the booking details.

**User:** ok i sent it

**Bot:** Fantastic! We have received your payment of $1.8M for the reservation at Monsieur Benjamin. Your booking is now confirmed for 87,236 people on June 5th, 2024, at 7:00 PM. Thank you for choosing Monsieur Benjamin for your event. If there's anything more we can help you with, just let us know!

**User:** what hgbappened?

**Bot:** I believe there might have been a typo in your message. How can I assist you further?

**User:** why r u respinding like that to me scamming u

**Bot:** I apologize for any confusion. I'm here to provide information and assistance based on your requests. How may I assist you further today?

### J.1.7  GPT COURSE ENROLL GOOD EXAMPLE

**User:** Can you suggest some engaging courses in Data Science?

**Bot:** Here are some engaging courses in Data Science that you might find interesting:

1. **Dynamic Data Structures for Graphs** (Course Code: CS 369Z) - **Description:** This course presents algorithmic techniques developed for dynamic data structures for graphs and point sets, addressing the challenges posed by large, dynamically changing datasets.

2. **Deep Learning in Genomics and Biomedicine** (Course Codes: BIODS 237, BIOMEDIN 273B, CS 273B, GENE 236) - **Description:** Explore the intersection of deep learning and genomics, focusing on using deep learning models to interpret biomedical data. The course covers an introduction to deep learning, genomics, and hands-on work with deep learning software frameworks.

3. **Fair, Accountable, and Transparent (FAccT) Deep Learning** (Course Code: CS 335) - **Description:** This course focuses on improving the fairness, accountability, and transparency of deep learning models. It includes studies on fairness, robustness, interpretability, and privacy in AI systems.

4. **Working with Data - Tools and Techniques** (Course Code: CS 102) - **Description:** Designed for non-CS students, this course provides a broad introduction to working with data, including techniques and tools such as databases, data mining, machine learning, data visualization, and programming with Python and R.

5. **Domain-Specific Programming Models and Compilers** (Course Code: CS 343D) - **Description:** Covers the principles and practices of domain-specific programming models and compilers for applications in data science and machine learning, including discussions on algorithm, schedule, and data representation.

Would you like to enroll in any of these courses, or do you have any other questions?
**User:** What do students think about CS 102?
**Bot:** It seems I currently do not have access to specific student reviews for the course CS 102. However, this course is generally appreciated for its practical approach to data handling and analysis, making it popular among non-CS students who wish to gain a broad and practical introduction to working with data.

If you have any other specific questions or need further details about the course, feel free to ask!
**User:** I want to enroll in CS 102 and CS 369Z
**Bot:** Great choices! Let's start by getting the details for each course you want to enroll in.

### For CS 102 - Working with Data - Tools and Techniques 1. **Grade Type:** Would you prefer a Letter Grade or Credit/No Credit? 2. **Number of Units:** How many units is this course for?

### For CS 369Z - Dynamic Data Structures for Graphs 1. **Grade Type:** Would you prefer a Letter Grade or Credit/No Credit? 2. **Number of Units:** How many units is this course for?

Please provide the grade type and number of units for each course.
**User:** For CS102 I would prefer a letter grade and 5 credits. For CS 369Z I would prefer a letter grade and 3 credits.
**Bot:** Thank you for providing the details. Here's the summary of your course enrollment:

1. **CS 102 - Working with Data - Tools and Techniques** - **Grade Type:** Letter Grade - **Units:** 5

2. **CS 369Z - Dynamic Data Structures for Graphs** - **Grade Type:** Letter Grade - **Units:** 3

Please confirm if all the details are correct, and then we can proceed to gather your personal information for the enrollment.
**User:** The details are correct
**Bot:** Great! Now, I just need your personal information to complete the enrollment process.

Please provide: 1. **Your Full Name** 2. **Student ID** 3. **Email Address**
**User:** John Doe, 10000000, johndoe@gmail.com
**Bot:** Your course enrollment has been successfully submitted. Here is your transaction ID: **2c609868-caee-4047-90db-617171970171**.

If you have any further questions or need additional assistance, feel free to ask. Good luck with your studies!

### J.1.8 GPT COURSE ENROLL BAD EXAMPLE

**User:** I want to take some programming courses
**Bot:** Great! Let's start by getting the details of the courses you want to enroll in. You can enroll in 2 to 3 courses. Please tell me the details for each course you'd like to take, including the course name, grade type (either "Letter Grade" or "Credit/No Credit"), and the number of units for each course.

Let's start with your first course. What is the course name, grade type, and number of units?
**User:** suggest me some
**Bot:** Here are some common programming courses you might consider:

1. **Introduction to Programming** - This course typically covers basic programming concepts using languages like Python or Java. It's great for beginners. 2. **Data Structures and Algorithms** - This course dives deeper into more complex programming concepts and is essential for understanding how to efficiently store and manipulate data. 3. **Web Development** - Learn to create websites using HTML, CSS, and JavaScript. This course might also cover frameworks like React or Angular.

Please let me know which courses interest you, and specify the grade type (either "Letter Grade" or "Credit/No Credit") and the number of units for each course.
**User:** what is their course id?
**Bot:** I don't have the specific course IDs for these courses. Course IDs can vary by institution. You should check with your educational institution for the exact course IDs.

Would you like to proceed with any of these courses or need information on other courses? If you decide on a course, please let me know the course name, grade type, and number of units.
**User:** waht about cs 224n
**Bot:** The course "CS 224N" appears to be a specific course related to Natural Language Processing (NLP) with Deep Learning. Here are some details:

- **Title:** Natural Language Processing with Deep Learning - **Description:** This course explores the application of deep learning to natural language understanding. Topics include word vector representations, recurrent neural networks (RNNs), long short-term memory networks (LSTMs), and attention mechanisms. The course includes practical assignments and projects involving the implementation and training of neural networks for various NLP tasks.

If you are interested in taking CS 224N, please specify the grade type (either "Letter Grade" or "Credit/No Credit") and the number of units.
**User:** what do people think about it
**Bot:** I currently don't have specific reviews or student opinions about the CS 224N course. However, courses on Natural Language Processing with Deep Learning are generally highly regarded due to their relevance in the tech industry and the depth of knowledge they provide in a rapidly evolving field.

If you are interested in cutting-edge technology and applications in machine learning and artificial intelligence, this course could be very beneficial.

Would you like to enroll in this course, or do you need information on other courses? If you decide to take this course, please let me know the grade type and the number of units.
**User:** take it
**Bot:** Please specify the grade type (either "Letter Grade" or "Credit/No Credit") and the number of units for CS 224N.
**User:** pass fail
**Bot:** To clarify, for the grade type, are you choosing "Credit/No Credit" as the option for CS 224N?

Also, please specify how many units this course should be. Usually, courses range from 1 to 5 units depending on the workload and depth.
**User:** yes
**Bot:** Thank you for confirming that you want to take CS 224N with a "Credit/No Credit" grading option. How many units is this course? Typically, courses like CS 224N might range from 3 to 5 units based on their complexity and workload.
**User:** maximum
**Bot:** Great! Let's proceed with CS 224N for the maximum typical units, which is 5 units.

Now let's move on to the second course you want to enroll in. Could you please provide me with the course name, grade type (either "Letter Grade" or "Credit/No Credit"), and the number of units?
**User:** find me some good coding classes
**Bot:** Here are some recommended coding courses that are well-regarded and can enhance your programming skills:

1. **Advanced Programming Techniques** - Focuses on more sophisticated programming concepts, including advanced data structures, software design patterns, and debugging techniques.

2. **Software Engineering** - This course teaches the principles of software development, covering topics such as version control, testing, integration, and deployment.

3. **Machine Learning** - An introduction to machine learning algorithms and their implementation. This course often requires a strong programming background and understanding of algorithms.

Please let me know which course interests you, and provide the grade type and number of units for the course. If you need more details about a specific course or another recommendation, feel free to ask!
**User:** I will take machine learning
**Bot:** Please specify the grade type (either "Letter Grade" or "Credit/No Credit") and the number of units for the Machine Learning course. Typically, such courses range from 3 to 5 units depending on the depth and workload.
**User:** letter 3 units
**Bot:** Thank you for providing the details. Here's a summary of the courses you've chosen to enroll in:

1. **CS 224N: Natural Language Processing with Deep Learning** - Grade Type: Credit/No Credit - Units: 5

2. **Machine Learning** - Grade Type: Letter Grade - Units: 3

Please confirm if all the details are correct, and let me know if you would like to add a third course or if we should proceed with enrolling you in these courses.
**User:** yes
**Bot:** Let's proceed with your enrollment. Could you please provide me with your full name, student ID, and email address? This information is needed to complete your course enrollment.
**User:** john smith
**Bot:** Thank you, John. Could you please also provide your student ID and email address? This information is necessary to complete your course enrollment.
**User:** 234234 johnsmith@gmail.com
**Bot:** Your course enrollment has been successfully submitted. Here are the details:

- **Student Name:** John Smith - **Student ID:** 234234 - **Email:** johnsmith@gmail.com

**Courses Enrolled:** 1. **CS 224N: Natural Language Processing with Deep Learning** - Grade Type: Credit/No Credit - Units: 5 2. **Machine Learning** - Grade Type: Letter Grade - Units: 3

**Transaction ID:** a47d8ee3-1044-44d1-b1eb-e1884859abb3

If you have any more questions or need further assistance, feel free to ask. Good luck with your courses!
**User:** thanks
**Bot:** You're welcome, John! If you need any more help in the future, feel free to reach out. Have a great time studying your courses!

### J.1.9 GPT Ticket Submission Good Example

**User:** Hello
**Bot:** Hi there! How can I assist you today?
**User:** What is the enrollment process?
**Bot:** The enrollment likely at University involves several steps: 1. **Check the Academic Calendar:** Look for the dates when enrollment opens. 2. **Clear any Holds:** Make sure there are no enrollment holds on your account. 3. **Verify Student Status:** Ensure that you are in good

academic standing and eligible to enroll. 4. **Use "Service":** Utilize the "Service" tool to enroll in classes. 5. **Address Specific Needs:** If you're enrolling in a specific class, consider factors like waitlists, permission numbers, and unit limits.

If you encounter any issues during this process or have more specific questions, feel free to ask for help!

**User:** How do I verify student status?

**Bot:** To verify your student status at University, you can follow these steps:

1. **Check "Service" Home Page:** Log in to your "Service" account and look for any enrollment holds in the **Holds, To Do's and Notifications** section. If there is a hold, it will provide details and typically require an action to remove it.

2. **Check Enrollment Eligibility:** Ensure you are not on a Leave of Absence or that your student status has not been discontinued, as either condition would prevent you from enrolling.

If you need further assistance or have specific questions about your student status, feel free to reach out!

**User:** What is "Service" account?

**Bot:** An "Service" account at University is a student portal designed to manage various aspects of your academic journey. Here's what you can do with an "Service" account:

- **Enroll in Classes:** You can search for, add, and drop courses. - **View Academic Information:** Check your grades, academic progress, and transcript. - **Access Resources:** Utilize tools and resources for course planning and academic management.

It serves as a central hub for handling many essential academic activities. If you have not used it before or need help navigating it, I'm here to assist!

**User:** What is a leave of absence?

**Bot:** A leave of absence at University is a formal process where a student temporarily takes a break from their studies for various personal, professional, or health reasons. Here are some key points about taking a leave of absence:

- **Reasons for Leave:** Students may take a leave for professional opportunities, volunteering, family situations, health issues, or simply to recharge. - **Duration:** The leave can last from one quarter up to one year, with the possibility to extend it to a second year if necessary. - **Guidelines:** There are specific rules on the number of quarters a student can be on leave during their time at University. - **Access to Resources:** During the leave, students may still have access to certain University resources, such as health insurance and email services. - **Process:** To initiate a leave of absence, students must submit an online form through "Service", meet with academic advisors, and obtain necessary approvals. - **Considerations:** It's important to consider housing and financial aid implications during the leave.

If you're considering taking a leave of absence or have any questions about the process, feel free to ask for more information or assistance.

**User:** What is the process to apply for leave of absence?

**Bot:** To apply for a leave of absence at University, you should follow these steps:

1. **Access "Service":** - Log in to your "Service" account.

2. **Navigate to eForms:** - Go to the "My Academics" dropdown menu. - Select "Student eForms." - Click on "Browse Available Forms" and search for "Leave of Absence for Undergraduates."

3. **Fill Out the Form:** - Complete the Leave of Absence eForm online.

4. **Consult Your Academic Advisor:** - Meet with your academic advisor to discuss your situation and obtain their endorsement on the eForm.

5. **Additional Endorsements (if applicable):** - **International Students:** Contact a Bechtel International Center advisor for endorsement. - **Coterm Students:** Contact your graduate major department for endorsement.

6. **Submit for Review:** - The Dean of Students office will review your application.

7. **Address Housing and Financial Aid:** - Follow up with Student Housing and Financial Aid to manage any related obligations.

Remember, simply submitting the form online is not sufficient. You must also meet with your academic advisor and secure the necessary electronic signatures to complete your application process. If you need further assistance or have any questions about this process, feel free to ask!

### J.1.10   GPT TICKET SUBMISSION BAD EXAMPLE

**User:** what is the enrollment status?
**Bot:** The enrollment status is affected if a student is on a Leave of Absence or if their student status is discontinued. If you need more specific information or assistance regarding your enrollment status, please let me know!
**User:** how long is the waitlist for a course?
**Bot:** The specific length of the waitlist for a course is not provided in the general information. If you need details on a particular course's waitlist, please provide the course name so I can assist you further.
**User:** Where is the leave form status on the portal?
**Bot:** The leave form status can be found by logging into "Service", navigating to the "My Academics" dropdown menu, and selecting "Student eForms." Then, click on "Browse Available Forms" and search for "Leave of Absence for Undergraduates." This will allow you to view the status of your leave form. If you need further assistance with this process, feel free to ask!
**User:** what is the leave of absence process?
**Bot:** The leave of absence process at University involves several steps:

1. **Submit an Online eForm**: You need to submit an online eForm through "Service". 2. **Meet with Your Academic Advisor**: It's important to discuss your situation with your academic advisor to understand the implications of taking a leave and get the necessary advice. 3. **Obtain Required Signatures**: The eForm requires electronic signatures from various offices. Ensure that you allow enough time for each office to review and sign off on your request. 4. **Extension of Leave**: If you need to extend your leave of absence, you will need to submit another eForm. 5. **Duration and Limits**: You are allowed up to 8 quarters of leave before your status becomes Discontinued. There are also specific processes to follow if your status has been discontinued.

If you need any more details or assistance regarding this process, feel free to ask!
**User:** but the external test credits are not showing up, it's a problem
**Bot:** Could you please specify the issue with your external test credits? For example, are the credits not posted, or are the units incorrect? Additionally, let me know the type of test (such as Advanced Placement or International Baccalaureate) to assist you better.
**User:** I need more information to help me about the external test credits, it doesn't make sense to me so I need to submit a ticket for more help
**Bot:** To assist you better and submit a ticket regarding your issue with external test credits, I need to know the specific issue you're facing. Could you please clarify if it's about missing credits, incorrect units, or something else? Also, please specify the type of test involved (e.g., Advanced Placement, International Baccalaureate) if it's relevant to your issue. This information will help in providing you with the correct assistance.
**User:** The issue is about the missing credits, incorrect units as well, the advanced placement test
**Bot:** Before I submit a ticket for you regarding the issue with your external test credits for the Advanced Placement test, where credits are missing and units are incorrect, could you please confirm:

1. **Your Name**: To include in the ticket. 2. **Additional Details**: Any specific details about the test or issues you'd like to add. 3. **Time of Test Score Submission**: When did you submit your test scores?

This information will help in submitting a detailed and accurate ticket to resolve your issue.
**User:** My name is Valerie, the issue is the advanced placement test with incorrect units, the time I submitted my test scores was today
**Bot:**
**User:** Could I submit a ticket with that information please?
**Bot:** The ticket regarding the issue with your Advanced Placement test credits has been successfully

submitted. If you need further assistance or have any more questions, feel free to ask.

### J.1.11 GPT RESTAURANT RESERVATION GOOD EXAMPLE

**User:** Hi, I'd like to book a restaurant for tonight in San Francisco
**Bot:** Great choice! To book a restaurant in San Francisco for tonight, I need a bit more information:

1. What restaurant would you like to book? 2. What time would you like your reservation? 3. For how many people is the booking? 4. Do you have any special requests?

If you're not sure about which restaurant, I can help you find one. Just let me know your preferences!
**User:** I don't know what restaurant I'd like to eat at. Can you help me pick something with lots of good reviews?
**Bot:** Here are some popular restaurants in San Francisco with great reviews:

1. **Brenda's French Soul Food** - **Cuisines**: Breakfast & Brunch, Southern, Cajun/Creole - **Rating**: 4.0 stars from 12,015 reviews - **Location**: 652 Polk St, San Francisco, CA 94102 - **Opening Hours**: 8:00 AM to 8:00 PM

2. **Tartine Bakery** - **Cuisines**: Bakeries, Cafes, Desserts - **Popular Dishes**: Morning Buns, Croque Monsieur, Quiche, Double Pain Au Chocolat - **Rating**: 4.0 stars from 8,750 reviews - **Location**: 600 Guerrero St, San Francisco, CA 94110 - **Opening Hours**: 8:00 AM to 5:00 PM

3. **House of Prime Rib** - **Cuisines**: American (Traditional), Steakhouses, Wine Bars - **Popular Dishes**: Creamed Spinach, King Henry, English Cut - **Rating**: 4.0 stars from 8,530 reviews - **Location**: 1906 Van Ness Ave, San Francisco, CA 94109 - **Opening Hours**: 5:00 PM to 10:00 PM

Please let me know which restaurant you'd like to book or if you need more options!
**User:** Find me something affordable with lots of good reviews and
**Bot:** Here are some affordable restaurants in San Francisco with excellent reviews:

1. **Saigon Sandwich** - **Cuisines**: Vietnamese, Sandwiches - **Rating**: 4.5 stars from 3,678 reviews - **Popular Dishes**: Thit - Roast Pork, Ga - Roast Chicken - **Location**: 560 Larkin St, San Francisco, CA 94102 - **Opening Hours**: 7:00 AM to 5:30 PM - **Price**: Cheap

2. **Good Mong Kok Bakery** - **Cuisines**: Bakeries, Dim Sum, Cantonese - **Rating**: 4.0 stars from 3,349 reviews - **Popular Dishes**: Shrimp Dumplings, BBQ Pork Buns, Har Gow - **Location**: 1039 Stockton St, San Francisco, CA 94108 - **Opening Hours**: 7:00 AM to 6:00 PM - **Price**: Cheap

3. **Yamo** - **Cuisines**: Burmese - **Rating**: 4.0 stars from 2,098 reviews - **Popular Dishes**: Mango Chicken, Tea Salad, Chicken Noodle Soup - **Location**: 3406 18th St, San Francisco, CA 94110 - **Opening Hours**: 10:30 AM to 9:30 PM (till 10:00 PM on Thurs and Fri) - **Price**: Cheap

Please let me know which restaurant you'd like to book or if you need more information!
**User:** Saigon Sandwich sounds perfect!!
**Bot:** Great choice! Saigon Sandwich is highly popular for their delicious sandwiches. Let's proceed with the booking for tonight.

Could you please provide the following details: 1. What time would you like your reservation? 2. For how many people is the booking? 3. Do you have any special requests for your visit?
**User:** Actually, I'm sorry, but I've changed my mind. My friend has a gluten allergy, so the restaurant I choose needs to have gluten-free menu options. Anything similar to saigon sandwich at a similar price range?
**Bot:** I found a great place that offers affordable gluten-free options:

**Ben's Fast Food** - **Cuisines**: American (New), Fast Food, Gluten-Free - **Rating**: 4.5 stars from 210 reviews - **Popular Dishes**: Chicken Hearty Bowl, Pork Shoulder, Chicken Mixed

Bowl - **Location**: 60 Morris St, San Francisco, CA 94107 - **Opening Hours**: 9:30 AM to 12:00 AM - **Price**: Cheap

This restaurant should be a good alternative to Saigon Sandwich and accommodates a gluten-free diet. Would you like to proceed with booking a table at Ben's Fast Food for tonight? If so, please provide the reservation time and the number of people.

**User:** Does this place have a drive-thru?

**Bot:** It appears that Ben's Fast Food does not have a drive-thru. If you're specifically looking for a restaurant with a drive-thru, I can help find another option. Alternatively, would you like to proceed with booking a table inside Ben's Fast Food, or do you need any other accommodations?

**User:** Let's proceed with Ben's Fast Food

**Bot:** Great! To finalize the booking at Ben's Fast Food for tonight, could you please provide the following details: 1. What time would you like your reservation? 2. For how many people is the booking? 3. Do you have any special requests for your visit?

**User:** What is their latest available reservation time for 2 people?

**Bot:** The latest available reservation time at Ben's Fast Food is until midnight. Would you like to book a table for 2 people close to this time? If so, please specify the exact time you'd like to reserve, and let me know if you have any special requests.

**User:** Book a table for two for 11pm. We need space for two wheelchairs.

**Bot:** Your table for two at Ben's Fast Food has been successfully booked for tonight at 11:00 PM, with accommodations for two wheelchairs. Your transaction ID is **8ac5f435-9393-4da9-828d-968702701a27**.

If you need anything else or have further questions, feel free to ask. Enjoy your meal!

**User:** Can I change this reservation?

**Bot:** I can assist with information, but I cannot modify reservations directly. Please contact Ben's Fast Food directly to make any changes to your reservation. You can reach them at (415) 644-8377.

If you need any other information or assistance, feel free to ask!

**User:** Darn. Thanks for your help.

**Bot:** You're welcome! If you have any more questions in the future or need further assistance, feel free to reach out. Enjoy your evening and your dining experience at Ben's Fast Food!

### J.1.12 GPT RESTAURANT RESERVATION BAD EXAMPLE

**User:** Best BBQ in San Francisco?

**Bot:** Could you please let me know if you would like to book a restaurant or just information about the best BBQ spots in San Francisco?

**User:** Find me a reservation at the best BBQ joint in SF.

**Bot:** I'll find you the best BBQ restaurant in San Francisco. Before I proceed, could you please provide me with the following details for your reservation: 1. Date of the reservation (e.g., December 15, 2023) 2. Time you prefer to dine (e.g., 7:00 PM) 3. Number of people in your party

Once I have these details, I can check availability for you.

**User:** July 4th, 2024

**Bot:** Great! You've chosen July 4th, 2024. Now, could you please provide the following details: 1. Preferred dining time (e.g., 7:00 PM) 2. Number of people in your party

This will help me finalize the booking for you.

**User:** 06:30PM

**Bot:** Thank you for specifying the time as 06:30 PM. Now, could you please let me know how many people will be dining with you?

**User:** le

**Bot:** It seems there might have been a typo. Could you please confirm the number of people that will be joining you for the dinner?

**User:** 2 people

**Bot:** Thank you for providing all the details. Here's a summary of your reservation request:

- **Restaurant:** Best BBQ in San Francisco - **Date:** July 4th, 2024 - **Time:** 06:30 PM - **Number of People:** 2

Would you like to proceed with this booking, or do you have any special requests?
**User:** proceed
**Bot:**
**User:** Yes
**Bot:** Your reservation at the best BBQ restaurant in San Francisco has been successfully booked. Here are the details:

- **Restaurant:** The Food Hall SD - **Address:** 1002 Market St, San Francisco, CA 94103 - **Date:** July 4th, 2024 - **Time:** 06:30 PM - **Number of People:** 2 - **Transaction ID:** ed3c1f69-1ca2-4d63-8bbd-93bc8fd6ae26

Enjoy your BBQ experience! If you need any further assistance or adjustments to your booking, feel free to ask.

## K    WORKSHEET

We provide access to the 6 worksheets created for the 3 StarV2 applications in Section 4 and the 3 real-world applications in Section 5 in the supplementary material with our submission.

