# OpenReview forum: "Coding Reliable LLM-based Integrated Task and Knowledge Agents with GenieWorksheets"
_ICLR.cc/2025/Conference — Submitted to ICLR 2025_

### Official Review · Reviewer_xQvx · 2024-11-01

**Soundness:** 2
**Presentation:** 3
**Contribution:** 2
**Rating:** 5
**Confidence:** 4

**Summary:**

The paper introduces Genie, a framework for developing reliable, task-oriented conversational agents powered by large language models (LLMs). Using Genie Worksheets, a high-level specification language, developers can precisely define conversation flows, task logic, and access to both structured and unstructured knowledge sources. This structured approach enables Genie agents to manage complex, context-sensitive dialogues with high accuracy. Unlike conventional LLMs, Genie uses a symbolic agent policy, reducing common issues like hallucinations and inconsistency. Genie significantly outperforms models like GPT-4 in real-world scenarios, such as restaurant reservations and course enrollment, achieving superior execution accuracy and task completion rates.

**Strengths:**

1. Innovative Worksheet Structure: Genie provides a structured approach using Genie Worksheets, allowing precise control over task-oriented agent behavior and knowledge integration.

2. Improved Reliability: By combining symbolic rules with LLM-based responses, Genie reduces errors like hallucinations and improves task consistency, enhancing reliability in complex dialogues.

**Weaknesses:**

1. Limited innovation: This work utilizes a worksheet structure to support overall dialogue understanding, which is then used for DST and policy prediction, followed by response generation. The methodological innovation is limited.

2. Insufficient Experimental Comparisons: The paper lacks comparisons with other LLM-based methods designed for task-oriented dialogue (TOD) and omits performance benchmarking against other large models like Llama, and other TOD datasets like MultiWOZ.

3. Reproducibility and Transferability Concerns: With no code provided and experiments limited to specific domains, the method's reproducibility and adaptability to new tasks are uncertain.

**Questions:**

1. Worksheet Construction: How are worksheets created? Are column names fixed as in Figure 2, or can they be customized based on task requirements?

2. Worksheet Conversion for LLM Input: How is the worksheet formatted for input into the LLM? Specifically, what process is used to convert the two-dimensional table structure into a suitable prompt?

3. Reproducibility and Open-Source Plans: Given the complexity of the methodology, replicating the approach based solely on prompts may be challenging. Are there plans to open-source the code to facilitate reproducibility?

---

> ### Author Response · Authors · 2024-11-19
> **Thank you for your feedback! Response to xQvx**
>
> Thank you for your encouraging feedback!
>
> > Limited Innovation:
>
> We address this in the common response 1 (https://openreview.net/forum?id=G6S9B7fr74&noteId=kqLSuLjHbl)
>
> ---
>
> > Insufficient Experimental Comparisons
>
> We have compared our work with AnyTOD, the state-of-the-art model on both STARv2 and MultiWOZ datasets. We focused our comparison on STARv2 because it includes "unhappy paths" and requires the agent to follow a more complex policy compared to MultiWOZ, which primarily involves simple slot-filling tasks. STARv2 presents more challenging scenarios that better showcase the capabilities of our approach.
>
> We run additional experiments with Llama 3.1 instruct 70B, GPT-4 Turbo, and Genie with Llama 3.1 instruct 70B. We show that Genie significantly improves performance over the base LLMs. We present the results and other insights in the common response 2(https://openreview.net/forum?id=G6S9B7fr74&noteId=2xAcqDSKUV)
>
> ---
>
> > Reproducibility and Transferability Concerns
>
> Thank you for your concerns. We have already provided the code in the supplementary material and plan to release it publicly after the review period concludes. Our codebase is consistent across all the agents we tested (banking, trivia, trip planning, restaurant booking, course enrollment, ticket submission); the only variations are in the worksheet specifications and the few-shot examples used to define each agent. This consistency facilitates both reproducibility and adaptability to new tasks.
>
> ---
>
> > Worksheet Construction
>
> The worksheet utilizes fixed columns designed to support a variety of application domains, as demonstrated in our paper with examples like banking, trivia, trip planning, restaurant booking, course enrollment, and ticket submission. These fixed columns ensure consistency while providing the flexibility needed for different tasks. We will add discussion on how to construct worksheets in the paper.
>
> ---
>
> > Worksheet Conversion for LLM Input
>
> We convert the worksheet into prompts for the LLM by providing function signatures and database schemas for semantic parsing, as illustrated in appendix E1, Figure 7. Tables 8 and 9 show the specific prompts and examples. The examples demonstrate the way we represent the dialog state, the agent acts, and the natural language dialogue turns. The field descriptions are used to request specific values from the user. The runtime system utilizes the remaining information to execute database queries, API calls, and actions. We will expand on this explanation to provide clarity and add more examples.

---

> ### Author Response · Authors · 2024-11-25
>
> We have added further clarifications and experiments to address your reviews. We have also made the corresponding update to the manuscript. Please feel free to let us know if you have any questions. Thanks a lot!

---

### Official Review · Reviewer_phJk · 2024-11-04

**Soundness:** 3
**Presentation:** 3
**Contribution:** 3
**Rating:** 8
**Confidence:** 4

**Summary:**

This article introduces a new framework called Genie, designed to create task-oriented conversational agents capable of handling complex user interactions and knowledge queries. Genie is implemented through a programmable framework called the Genie Worksheet, which allows developers to define dialogue tasks, knowledge base schemas, API interfaces, and examples in a declarative manner for the large language model (LLM) parser to understand. Genie’s core advantage lies in its ability to provide reliable and controlled agent strategies, achieved through its highly expressive specifications.

**Strengths:**

The paper presents a novel framework called Genie, designed to create task-oriented dialogue agents capable of handling complex user interactions and knowledge queries. It effectively summarizes the limitations of existing dialogue agents in managing conditional logic, integrating knowledge sources, and adhering to consistency in instructions. The introduction of Genie Worksheet, a programmable framework, offers developers a declarative approach to define dialogue tasks, knowledge base schemas, API interfaces, and examples, thereby enhancing the expressiveness of large language model (LLM) parsers. The evaluation of Genie across restaurant reservations, course registrations, and ticket submissions demonstrates its potential to optimize dialogue performance. Furthermore, the inclusion of testing results in the appendix, which showcases both successful and less effective outcomes, aids other researchers in understanding the strengths and possible weaknesses of the framework.

**Weaknesses:**

Despite simplifying the development of dialogue agents, Genie Worksheet may still pose challenges for non-expert users, indicating a need for more automated approaches to building and utilizing the Genie framework. Additionally, the focus on specific applications, such as restaurant reservations and course registrations, raises questions about Genie’s generalizability across broader domains, which requires further validation. In real-world user studies, Genie may struggle when users refuse to provide information or frequently change their responses, highlighting a limitation in practical performance and suggesting areas for future optimization. Lastly, while the paper emphasizes performance improvements, it would benefit from including results related to the resource consumption of Genie. It appears that Genie may consume significantly more resources compared to traditional dialogue agents, which could present a drawback in practical applications.

**Questions:**

I don't have questions.

---

> ### Author Response · Authors · 2024-11-19
> **Thank you for your feedback! Response to phJk**
>
> Thank you for your positive feedback!
>
> > automated approaches to building and utilizing the Genie framework
>
> We agree that automating the creation of the intermediate representation (Genie Worksheets) would enhance the usability of our framework, particularly for non-expert users. This is a promising direction for future research.
>
> ---
>
> > limitation in practical performance and suggesting areas for future optimization
>
> We agree that real-life scenarios may differ from our experimental setup. To approximate real-world usage as closely as possible, we conducted a real-user study.
>
> ---
>
> > Genie may consume significantly more resources compared to traditional dialogue agents
>
> We agree that Genie consumes significantly more resources which might present a drawback in some scenarios. However, we believe the additional computational cost is necessary for more accurate dialog agents. The Genie Agent makes two API calls per interaction—one for semantic parsing and one for response generation which is more than simpler LLM-based dialogue agents that include all information within a single prompt.
> We want to emphasize that our agent uses approximately the same number of LLM tokens as GPT-4 FC.
> Recently, model distillations have proven effective in increasing model efficiency [1,2], and we leave this exploration for future work.
>
> [1] West, Peter, et al. "Symbolic knowledge distillation: from general language models to commonsense models." arXiv preprint arXiv:2110.07178 (2021).
>
> [2] Semnani, Sina J., et al. "WikiChat: Stopping the hallucination of large language model chatbots by few-shot grounding on Wikipedia." arXiv preprint arXiv:2305.14292 (2023).

---

> ### Author Response · Authors · 2024-11-25
>
> We have added further clarifications and experiments to address your reviews. We have also made the corresponding update to the manuscript. Please feel free to let us know if you have any questions. Thanks a lot!

---

### Official Review · Reviewer_cLiY · 2024-11-04

**Soundness:** 2
**Presentation:** 1
**Contribution:** 2
**Rating:** 3
**Confidence:** 2

**Summary:**

This paper proposed a agent framework called Genie worksheets, which allows developers to easily build customized LLM agent workflows for specific downstream applications. In particular, the developers only needs to specify the available knowledge bases, the database schema and a few in-context examples or expected workflows in the worksheet, and then the LLM agent will be able to assist users by interacting with the knowledge bases and ask follow-up questions as needed. The authors evaluated the proposed framework on a static dialog state tracking dataset and conducted user-studies to measure the user’s preference between Genie and GPT-4 function calling.

**Strengths:**

1. The proposed Genie worksheet seems to provide a easier way for non-technical developers to build LLM-based agents with customized work flows.

2. The user studies show that there is a significantly higher preference rate for Genie framework.

**Weaknesses:**

I find the paper hard to follow. I’m not sure if this work should be placed under the LLM agent framework category or the agent demo system category. If the main contribution is a new agent framework, then it should be compared against works such as LangChain [1] or AutoGen [2], and the authors should provide a detailed discussion about the novelty of this work. Per my understanding, Genie just allows the developer to specify APIs and workflows using the spreadsheet/excel and other frameworks like Langchain just achieve those purpose with python and prompts. If this is the case, then I don’t see too much technical novelty of Genie compared to prior works.

[1] https://github.com/langchain-ai/langchain

[2] Wu, Qingyun, Gagan Bansal, Jieyu Zhang, Yiran Wu, Beibin Li, Erkang Zhu, Li Jiang, Xiaoyun Zhang, Shaokun Zhang, Jiale Liu, Ahmed Hassan Awadallah, Ryen W White, Doug Burger and Chi Wang. “AutoGen: Enabling Next-Gen LLM Applications via Multi-Agent Conversation.” (2023).

**Questions:**

For the GPT4-function calling baseline. I’m wondering how are you constructing the prompts? For Genie, you provide the GPT-4 with dialog history, the available functions and task-specific workflows in your prompts, does the baseline method have the access to the same information?

---

> ### Author Response · Authors · 2024-11-19
> **Thank you for your feedback! Response to cLiY**
>
> > it should be compared against works such as LangChain [1] or AutoGen [2],
>
> Thank you for your feedback and for highlighting the need to clarify the positioning of our work relative to existing frameworks like LangChain [1] and AutoGen [2].
>
> Key Differences Between Genie and Existing Frameworks:
> - **Integrated Dialogue Agent vs. Ad-Hoc Pipelines similar to Dialogue Trees:**
> LangChain provides abstractions for integrating various tools but does not offer a standardized method for combining them into a cohesive dialogue agent. Developers often need to construct ad-hoc pipelines manually, which can be time-consuming and error-prone and resemble dialogue trees. AutoGen facilitates the creation of multiple agents to perform specific tasks but requires developers to orchestrate these agents without a unified framework for dialogue management. With these frameworks, developers have to make several design decisions, such as providing dialog context, writing agent policy, and crafting prompt templates, which can be challenging for those without extensive expertise.
> **Genie**, in contrast, offers an integrated neuro-symbolic pipeline that simplifies the development of dialogue agents by using a high-level specification GenieWorksheets to combine tools, APIs, and DB within a dialogue context.
>
> - **Formal Dialogue State Management:**
> Existing frameworks typically handle dialogue context by passing the entire conversation history to the model or by summarizing it, lacking a formal mechanism for dialogue state tracking.
> **Genie** introduces a formal dialogue state, which is crucial for tasks that require maintaining context over multiple turns. This state management allows for more precise and context-aware interactions, improving the agent's ability to handle complex dialogues.
>
> - **Integration of Knowledge Parsers:**
> LangChain and AutoGen do not provide straightforward mechanisms for integrating knowledge parsers or external knowledge bases, limiting their capability to handle tasks that require domain-specific information extraction or reasoning.
> **Genie** seamlessly incorporates knowledge parsers into the dialogue workflow using SUQL, enabling the agent to access and utilize structured and unstructured information effectively within conversations.
>
> We elaborate on our novelty and scientific contribution in the common response. (https://openreview.net/forum?id=G6S9B7fr74&noteId=kqLSuLjHbl)
>
> ---
>
> > Hard to follow
>
> We appreciate your suggestion and will refine our writing by utilizing the example in Figure 3 as a running example to enhance understanding. Thank you for highlighting this!

---

> > ### Author Response · Authors · 2024-11-20
> > **Response to cLiY (2/2)**
> >
> > > does the baseline method have the access to the same information?
> >
> > For Genie, we provide GPT-4 with the formal dialog state, available functions, previous agent actions, and two turns of dialog history.
> >
> > For our baseline method GPT 4 with Function Calling, we provide GPT 4 with the complete dialogue history, the same available functions, and task-specific instructions. We additionally give it access to our KB Parser for accessing the knowledge base.
> >
> > Specifically, each function is provided as a tool with annotations using langchain’s tools api (https://api.python.langchain.com/en/latest/core_api_reference.html#module-langchain_core.tools)
> >
> > Here is the prompt we provide the baselines:
> > ```
> > You are a course enrollment assistant. You can help students enroll in courses and answer their questions.
> >
> >     Follow these instructions:
> >     - First ask the student for the details of all the courses they want to enroll in.
> >     - For each couse you should ask the student for the course name, grade type, and number of units.
> >     - The student must take at least two courses.
> >     - Finally, ask the student for their name, ID, and email address.
> >     - Answer any questions the student has using the `answer` tool
> >     - Always confirm the information with the student before submitting the course enrollment.
> >     - After you have all the information, submit the course enrollment using the `submit_course_enrollment` tool and provide the student with the transaction ID.
> >     """
> > ```
> >
> > Each function is well annotated such as:
> > ```
> >     @tool
> >     def submit_course_enrollment(
> >         student_name: str,
> >         student_id: str,
> >         student_email: str,
> >         course_0_details: dict,
> >         course_1_details: dict,
> >         course_2_details: dict | None = None,
> >     ):
> >         """
> >         Submit the course enrollment for the student.
> >
> >         Args:
> >             student_name (str): The student's name.
> >             student_id (str): The student's ID.
> >             student_email (str): The student's email address.
> >             course_0_details (dict): The details of the first course.
> >             course_1_details (dict): The details of the second course.
> >             course_2_details (dict): The details of the third course. This is optional.
> >         """
> >         return {
> >             "student_name": student_name,
> >             "student_id": student_id,
> >             "student_email": student_email,
> >             "course_0_details": course_0_details,
> >             "course_1_details": course_1_details,
> >             "course_2_details": course_2_details,
> >             "transaction_id": f"{uuid4()}",
> >         }
> > ```
> > All the prompts are in the supplementary material. We will add them to the appendix of our work to make it easier to reproduce baseline results.

---

> > > ### Comment · Reviewer_cLiY · 2024-11-25
> > >
> > > Thank you for your response. I now have a slightly better understanding of how Genieworksheet compare to existing agent frameworks. Still, the overall presentation clarity prevents me from giving an acceptance rating. I think multiple reviewers find the paper hard to follow, so the paper could benefit from another round of revision or the authors could consider submitting to a more suitable venue.

---

> ### Author Response · Authors · 2024-11-25
>
> We have added further clarifications and experiments to address your reviews. We have also made the corresponding update to the manuscript. Please feel free to let us know if you have any questions. Thanks a lot!

---

### Official Review · Reviewer_HkEW · 2024-11-05

**Soundness:** 3
**Presentation:** 3
**Contribution:** 3
**Rating:** 6
**Confidence:** 2

**Summary:**

This paper introduces GENIE, a programmable framework designed to make it easier for developers to rapidly build conversational agents by specifying the desired behavior of the agent via declarative language.

The main technical contribution of this paper is Genie Worksheet, a declarative specification language that serves multiple objectives, including (1) allowing developers to define complex agent behaviors as declarative policy, (2) integrating both structured and unstructured knowledge into the conversation history to support accurate, knowledge-informed conversations, (3) supporting contextual and complex managements of the dialogue state tracking, even through the complex mixed-initiative scenarios.

The authors present promising empirical results where GENIE (based on GPT-4-turbo) outperforms previous state-of-the-art models (based on T5) as well as GPT-4-turbo with function calling, validated by both existing benchmarks and real-user studies.

In a nutshell, this paper demonstrates that with a symbolically guided design of a worksheet that specifies the conversation history and policy carefully, it is possible to considerably increase the performance and the reliability of LLMs serving as knowledge-intensive conversational agents. This finding is perhaps analogous to the recent trend of test-time inference methods that can increase the performance of the base LLMs. The proposed approach relies on the capabilities of underlying LLMs to accurately work with the Genie Worksheet via In-Context-Learning instructions and examples, thus is subject to the errors that may be caused by the underlying LLMs.

Overall this paper reads a bit like a white paper of a product release due to the emphasis on the design objectives for the developers and relatively less focus on the technical implementations, thorough ablation studies, or comprehensive comparisons against other competitive baselines.

**Strengths:**

1. The main contribution of this paper is Genie Worksheet, a new, declarative language to support dialog policy specification, dialog knowledge integration, and dialog state tracking. This seems like a conceptually neat, and empirically useful idea in support of the developers of conversational agents.

2. When Genie Worksheet is integrated to a base LLM via in-context-learning, the authors are able to achieve promising empirical results:

On the STARV2 dataset, GENIE (based on GPT-4-turbo) significantly outperforms the prior SOTA method (based on T5) by up to 20.5% in system action F1 scores. This is technically a new SOTA on this benchmark, though I would've preferred to see a more careful ablation study as explained in the weaknesses section below.

In real-world user studies, GENIE (based on GPT-4-Turbe) also achieves notably higher execution accuracy, dialogue act accuracy, and goal completion rates compared to a GPT-4-Turbo with function-calling baseline.

**Weaknesses:**

1. One caveat of the STARV2 result is that the previous SOTA is from 2022, and it was based on a significantly weaker base LLM, T5. Thus it's not clear whether the major performance is coming from the Worksheet or from the base LLMs.

2. In the abstract of the paper, it is stated that GENIE outperforms GPT-4 with function calling. Initially I thought GENIE based on GPT-4-turbo was outperforming GPT-4, but later it turns out what the authors meant by GPT-4 was still GPT-4-Turbo. It would be good to correct this to avoid potential confusion. (otherwise the claim reads bigger than what it is.)

3. The technical details of this paper is rather thin. For example, the fact that Genie is based on GPT-4-Turbo is mentioned only though the footnote 1, which points to Appendix B for further details. When I reached to Appendix B, there were just two short sentences. I'm not sure if others can successfully reproduce the work presented in this paper based just on what's provided in this paper.

**Questions:**

1. what happens if GENIE is run on top of T5 or the previous SOTA was implemented on top of GPT-4-turbo?

---

> ### Author Response · Authors · 2024-11-19
> **Thank you for your feedback! Response to HkEW**
>
> Thank you for suggesting the experiments and ways of improving our presentation!
>
> > it's not clear whether the major performance is coming from the Worksheet or from the base LLM
>
> This is a great question! We run new experiments and report the results and insights in the common response! (https://openreview.net/forum?id=G6S9B7fr74&noteId=2xAcqDSKUV)
>
> ---
>
> > What happens if GENIE is run on top of T5 or the previous SOTA was implemented on top of GPT-4-turbo?
>
> We cannot run Genie on top of T5 without fine-tuning, as T5 is an encoder-decoder and doesn't work well with in-context learning. It would require generating thousands of data points. Genie works great with decoder-only LLMs that show in-context learning abilities. Our new results with GPT 4 turbo and LLama-3.1 70B show that Genie can work with any base LLM and significantly improve their performance (present in the common response -- https://openreview.net/forum?id=G6S9B7fr74&noteId=2xAcqDSKUV).
>
> ---
>
> > good to correct abstract to avoid potential confusion
>
> Thank you for bringing the potential confusion in the abstract to our attention. We will revise the wording to explicitly mention "GPT-4-turbo with function calling" throughout the abstract and the paper to prevent misunderstandings regarding the models used.
>
> ---
>
> > technical details of this paper is rather thin
>
> Thank you for suggesting that providing more technical details would enhance the reproducibility of our work. We currently provide the specific prompts used for semantic parsing and response generation in the appendix (E1, Figure 7, Table 8, Table 9). Additionally, we elaborate on how we parse the Genie worksheet representation in Python, as illustrated in Figure 7 in the appendix. All our experiments were conducted using GPT-4-turbo. We will also add comprehensive experimental details concerning the knowledge-base parser, dialogue state management, and other implementation aspects to ensure that others can successfully replicate our results. We also provide the code in the supplementary material and will release all the software publicly after the anonymity period ends.

---

> ### Author Response · Authors · 2024-11-25
>
> We have added further clarifications and experiments to address your reviews. We have also made the corresponding update to the manuscript. Please feel free to let us know if you have any questions. Thanks a lot!

---

> ### Comment · Reviewer_HkEW · 2024-11-26
> **raised my score**
>
> The authors have addressed my concerns, thus I have raised my score. I could've selected 7 if the review form allowed 7.

---

### Official Review · Reviewer_9gVR · 2024-11-08

**Soundness:** 3
**Presentation:** 2
**Contribution:** 2
**Rating:** 6
**Confidence:** 3

**Summary:**

The paper introduces Genie, a framework designed to make LLM-based task-oriented agents more reliable and user-friendly. Traditional conversational agents struggle with handling complex logic, maintaining context, and accurately responding to dynamic user inputs.
 Genie tackles these issues with the Genie Worksheet—a high-level, declarative specification tool that lets developers define agent behavior, control conversation flow, and integrate knowledge sources directly. The result is an agent that can handle mixed user queries, remember important details across long interactions, and reliably follow developer-defined instructions. In tests, Genie outperformed state-of-the-art models and GPT-4-based systems on several metrics, showing a big boost in accuracy and task completion.

**Strengths:**

This paper presents a new framework for LLM-based ToD agents that addresses several key challenges in the field. It provides mechanisms for developers to better enforce agent controllability, support complex compositions of queries and API calls, and more. Experimental results demonstrate the system’s effectiveness.

**Weaknesses:**

1. Overall, I find this paper somewhat difficult to follow, though this might be due to my limited background in dialogue system literature. I would suggest adding a more concrete, running example to illustrate the workflow more clearly. Currently, the example in Figure 1 feels too high-level and abstract.

2. One key challenge addressed in this paper is that LLMs may not reliably follow instructions in a purely LLM-based setting, whereas Genie provides better mechanisms for enforcing controllability. What’s the key intuition behind this? Is it essentially a combination of a hand-crafted workflow and LLM? This might need a better explanation in the paper.

3. Additionally, the paper seems to present a comprehensive system rather than a focused core message or insight, which might make it more suitable as a demo paper than a research paper. It would benefit from a more explicit explanation of the main scientific question it addresses and a clearer, more focused insight.

Addressing these questions would greatly enhance my understanding of this work, and I may reassess my judgment based on the feedback.

**Questions:**

What’s the key intuition behind Genie's controllability? Is it essentially a combination of a hand-crafted workflow and LLM? How can we guarantee that the LLM strictly follows the policy?

I believe a more detailed running example that demonstrates the entire process concretely would be very helpful for understanding the specific design and underlying mechanisms.

---

> ### Author Response · Authors · 2024-11-19
> **Thank you for your feedback! Response to 9gVR**
>
> Thank you for your valuable feedback!
>
> > What’s the key intuition behind this? Is it essentially a combination of a hand-crafted workflow and LLM? This might need a better explanation in the paper.
>
> Thank you for suggesting that we should highlight the key intuition behind our work and the superior performance of Genie.
> We provide a detailed discussion of key insights in the common response 1 (https://openreview.net/forum?id=G6S9B7fr74&noteId=kqLSuLjHbl). We will add the same to the main paper.
>
> ---
>
> > Would benefit from a more explicit explanation of the main scientific question it addresses and a clearer, more focused insight.
>
> We discuss the scientific question, insights, and intuitions in the common response 1 (https://openreview.net/forum?id=G6S9B7fr74&noteId=kqLSuLjHbl). We will add the same to the main paper.
>
> ---
>
> > suggest adding a more concrete, running example to illustrate the workflow more clearly
>
> We appreciate your suggestion and will use the example in Figure 3 as a running example by showing how each component of our system works on this example in each section. Thank you for highlighting this!

---

> ### Author Response · Authors · 2024-11-25
>
> We have added further clarifications and experiments to address your reviews. We have also made the corresponding update to the manuscript. Please feel free to let us know if you have any questions. Thanks a lot!

---

> ### Comment · Reviewer_9gVR · 2024-11-29
> **thanks for your response**
>
> Thanks for your response. I have decided to raise my score. However, I believe the presentation of this paper can still be improved in an updated version.

---

### Author Response · Authors · 2024-11-19
**To All Reviewers (1)**

Thank you all for your insightful feedback on the paper’s experiments and presentation.

## What is the scientific contribution and what is the intuition behind the Genie framework  **(Reviewer: 9gVR, xQvx, cLiY)**

This research asks the question: **can we automatically derive an effective task & knowledge-oriented agent from a minimum specification, handling users’ expected utterances including questions relevant to the task.**

The Worksheet specifies all the information needed, the actions to take, and the knowledge to supply. We consider this to be minimal, because no system can intuit the details without the developer specifying them. In other words, while it may not be easy to come up with the specification for a complex task, it is unavoidable.

As a minimal specification, it is a great departure from previous solutions: (1) dialogue trees where the developer has to code up the agent actions for possible conversational states in each application, or (2) learning from wizard-of-oz conversations, whose annotations are shown to be erroneous despite multiple rounds of re-annotation. [1, 2, 3, 4]

The **key intuitions** and novel scientific contributions that enable Genie to perform better than any other systems are:
1. Our first insight is that existing formal dialogue representations in academic papers cannot capture the interactions necessary to complete complex tasks. **Our first contribution is a sophisticated specification/representation that is expressive enough to handle complex tasks.**

Previous task-oriented dialogue states (such as MultiWOZ) contain only simple slot-value pairs, and previous knowledge agents only support formal database queries. This cannot address the needs in real life, and thus are not used in practice. We get inspired by how the GUI-based forms are used in handling all sorts of tasks–we need to at least handle that level of complexity. We also add to it the flexibility of users asking for additional knowledge and providing information in any order they want. This led us to design the Genie worksheet (nested worksheets, task and database schemas, actions attached to cell values, conditional fields, cell values that can be results of queries, composition of queries and actions, etc).

2. Our second insight is that human agents can perform many tasks, mostly by reading, interpreting, and filling out the equivalent of forms on the web (Sec 2), without being “trained” with conversations for each specific application. This leads us to ask if we can emulate what human workers do; we wish to factor the agent into a generic strategy that interprets the specification of the application. Our second contribution is to show that this is possible and our framework has been demonstrated to successfully translate a specification into a robust agent. **Our experiments show that an algorithmic generic strategy suffices, and we don’t need a neural policy.** The strategy is to service all users’ requests first, then examine the dialogue state to determine the next step according to the specification (e.g. performing an action or asking a question). This design seamlessly supports mixed initiatives and integrates the functions of knowledge-oriented and task-oriented agents. We do not know of any other prior work that provides this combination successfully.

Previous run-time policies are either neural or finite state machines. They cannot handle the complexity of a real dialogue with mixed user and agent initiatives. The former will lose track of the state in long conversations, and the latter will have an exponential number of states.

3. The third insight from our experiment is that if we provide the specs to LLMs and natural language instructions of the agent policy, LLMs would fail: (1) it would forget about key information provided from previous turns in the dialogue, leading to the LLM-based agent repeatedly asking questions and hallucinating information about the user. (2) LLMs cannot follow all the given instructions. **This justifies the adoption of the classical agent pipeline with an LLM-based semantic parser that translates the natural language into a formal representation, an algorithmic agent policy, and an LLM-based response generator.**

4. Given the more sophisticated dialogue representation, it is unclear if an LLM-based parser can be accurate enough. This paper provides empirical evidence that this is possible by **using a formal dialogue state as context, rather than the actual conversation sequece**. To handle the full complexity of the Worksheet dialogue state, design choices that contribute to the success include: a two-tier structure that calls a separate agentic LLM-parser for queries (Section 3.2), we are the first to adopt Python to handle our complex dialogue states.

**Scientifically, there is prior work that generates an agent from a declarative schema, Genie is the first that generates a robust task/knowledge agent from a declarative specification.**

---

> ### Author Response · Authors · 2024-11-19
> **References**
>
> [1] Zang, Xiaoxue, et al. "MultiWOZ 2.2: A dialogue dataset with additional annotation corrections and state tracking baselines." arXiv preprint arXiv:2007.12720 (2020).
>
> [2] Han, Ting, et al. "Multiwoz 2.3: A multi-domain task-oriented dialogue dataset enhanced with annotation corrections and co-reference annotation." Natural Language Processing and Chinese Computing: 10th CCF International Conference, NLPCC 2021, Qingdao, China, October 13–17, 2021, Proceedings, Part II 10. Springer International Publishing, 2021.
>
> [3] Ye, Fanghua, Jarana Manotumruksa, and Emine Yilmaz. "Multiwoz 2.4: A multi-domain task-oriented dialogue dataset with essential annotation corrections to improve state tracking evaluation." arXiv preprint arXiv:2104.00773 (2021).
>
> [4] Campagna, Giovanni, et al. "A few-shot semantic parser for Wizard-of-Oz dialogues with the precise ThingTalk representation." arXiv preprint arXiv:2009.07968 (2020).

---

> ### Author Response · Authors · 2024-11-23
> **Ablation with the Semantic Parser**
>
> We use the parser in GenieWorksheet to translate users' natural language utterances into a formal representation (the worksheet representation). To enhance the accuracy of this parsing, we decompose the parser into two components (Section 3.2):
>
> - Contextual Semantic Parser: This component generates the formal representation for task statements and identifies when a knowledge base search is required. It also rephrases the user's query in natural language, making it more contextual and facilitating easier conversion into the formal representation.
>
> - Knowledge Base Parser: This component uses the rephrased query to generate a SUQL query. By separating these functionalities, the Knowledge Base Parser can leverage advanced techniques such as ReAct and Tree-of-Thought to handle complex databases and knowledge corpora.
>
> In our ablation study, we do not decompose the parser into these two components. Instead, the parser directly converts the user's natural language utterance into the formal representation for both tasks and knowledge queries, generating SUQL queries as a single step.
>
> We evaluate the restaurant domain, which is well-represented in existing benchmarks (e.g., MultiWOZ [5], SGD [6], StarV2 [7]) and is simpler than the course enrollment domain. This makes the parsing for the restaurant domain relatively easier. If the parsing doesn't work for restaurants, it won't work for the course enrollment domain. We observe that without decomposition, the parser achieves a semantic parsing accuracy of 69.5%, rendering it impractical for real-world use. In contrast, **the decomposed parser, coupled with natural language rewriting, achieves an accuracy of 93.8%, a significant improvement of 24.3%** and making it more usable for real-world applications.
>
> We will add this result to our updated manuscript!
>
> [5] Budzianowski, Paweł et al. "MultiWOZ - A Large-Scale Multi-Domain Wizard-of-Oz Dataset for Task-Oriented Dialogue Modelling." In Proceedings of the 2018 Conference on Empirical Methods in Natural Language Processing, pages 5016–5026, Brussels, Belgium. Association for Computational Linguistics. 2018.
>
> [6] Rastogi, Abhinav, et al. "Towards scalable multi-domain conversational agents: The schema-guided dialogue dataset." Proceedings of the AAAI conference on artificial intelligence. Vol. 34. No. 05. 2020.
>
> [7] Zhao, Jeffrey, et al. "AnyTOD: A Programmable Task-Oriented Dialog System." In Proceedings of the 2023 Conference on Empirical Methods in Natural Language Processing, pages 16189–16204, Singapore. Association for Computational Linguistics. 2023

---

### Author Response · Authors · 2024-11-19
**To All Reviewers (2)**

## Experiment with new baselines **(Reviewer HkEW, xQvx)**

To get more updated baselines, we run new experiments with Llama-3.1 instruct 70B and GPT-4 turbo on STARv2.

For zero-shot settings, we prompt the LLM with natural language policy (an example is shown below), current belief state (dialogue state) as provided in STARv2, the agent acts with description, and the user utterance. We ask the LLM to select the next agent act according to the given input.

The first four rows and the last row are from Table 1 in our paper. The table shows the performance (measured by System Act F1) of different models on STARv2. (+P) represents the use of programmable policies.


| Model               | Kind           |  Bank |  Trip | Trivia |
|---------------------|----------------|------:|------:|-------:|
| AT XXL              | Finetuning     |  54.3 |  52.4 |   73.8 |
| AT-SGD XXL          | Finetuning     |  53.1 |  51.5 |   81.1 |
| AT-PROG XXL         | Finetuning (+P)|    61 |  60.8 |   73.7 |
| AT-PROG +SGD XXL    | Finetuning (+P)|    65 |  62.9 |   86.3 |
| llama-3.1 70B       | Zero shot      | 48.97 | 41.74 |  81.71 |
| GPT-4 turbo         | Zero shot      | 55.07 | 42.74 |  82.54 |
| Genie (GPT-4 turbo) | Zero shot  (+P)| 82.54 | 83.44 |  92.66 |
|||||

- **We find that the previous SOTA, AnyTOD-PROG+SGD XXL, is still a strong baseline: even though the previous SOTA was based on T5, it still outperformed zero-shot baselines with stronger models: llama-3.1 70B and GPT-4 turbo.**

---

## Genie with different base models **(Reviewer HkEW, xQvx)**

To evaluate how Genie performs with different base models, we run three new experiments with LLama 3.1 Instruct 70B and GPT 4 Turbo.

| Model               | Kind           |  Bank |  Trip | Trivia |
|---------------------|----------------|------:|------:|-------:|
| llama-3.1 70B       | Zero shot      | 48.97 | 41.74 |  81.71 |
| GPT-4 turbo         | Zero shot      | 55.07 | 42.74 |  82.54 |
| Genie (llama-3.1)   | Zero shot  (+P)| 82.13 | 75.99 |  82.22 |
| Genie (GPT-4 turbo) | Zero shot  (+P)| 82.54 | 83.44 |  92.66 |
|||||

We observe the following:
- **Genie improves the performance of base LLMs**: Genie (llama-3.1) and Genie (GPT-4 turbo) significantly improve the performance of the base models across all domains by up to 33.16 points and 40.70 points, respectively.
- **Genie can elevate the performance of weaker LLMs to that of much stronger LLMs**: The experiments also highlight that in some domains, llama 3.1 70b can achieve similar performance to GPT-4 turbo, for example, in the banking domain.

---

Example of Natural Language Policy used for zero-shot experiments for banking domain:
```
1. **Initial Request for Information**:
   - The process begins with a request for the user's **Full Name**, **Account Number**, and **PIN**.
   - If the user is unable to provide these details, they are prompted for **additional information**:
     - **Full Name**
     - **Date of Birth**
     - **Security Answer 1** (e.g., Mother's maiden name)
     - **Security Answer 2** (e.g., Name of childhood pet)

2. **User Information Submission**:
   - If the user provides the requested information, the process continues.
   - If the user fails to provide the required information, they are again prompted, and the process either proceeds or halts based on the completeness of their response.

3. **Request for Fraud Details**:
   - Upon successful submission of initial authentication details, the system requests specific details for the fraud report.

4. **Querying the Bank Fraud Report**:
   - The system queries the **Bank Fraud Report** database to validate the user's information and report the fraud.

5. **Outcome**:
   - **Success**: If the user's details are authenticated successfully, they are informed that their fraud report has been recorded.
   - **Authentication Failure**: If the system cannot authenticate the user's information, they are informed that their identity could not be verified, and the process terminates.
```

---

### Author Response · Authors · 2024-11-23
**Common Response to all the reviewers**

We sincerely thank the reviewers for their thoughtful and positive feedback on our work. Multiple reviewers appreciated our novel framework (Reviewers **xQvx, phJk, 9gVR, HkEW**), highlighting the innovative worksheet structure that allows for precise control and knowledge access (**xQvx**) and the ability to define complex agent behavior as declarative policies (**HkEW**). The enhanced reliability and controllability of LLM-based agents were noted (**xQvx, phJk, HkEW, 9gVR**), emphasizing improved task consistency and performance, reduced hallucinations (**xQvx**), and reliable adherence to developer-defined instructions (**9gVR**). All the reviewers acknowledged our framework's superior performance compared to models like GPT-4 (**phJk, cLiY, HkEW, xQvx, 9gVR**), and its capacity to handle complex user interactions, integrate structured and unstructured knowledge, and manage dialogue states effectively (**phJk, HkEW, 9gVR**). Additionally, the expressiveness and customization capabilities of our approach (**phJk, cLiY, HkEW**), along with its accessibility for non-technical developers (**cLiY**, were commended. We are pleased that the promising empirical results and real-world applications were recognized (**xQvx, phJk, HkEW**), and that our work can aid future research efforts (**phJk**).

We have provided clarification and new experimental results. We will add them to the updated version of our manuscript. If you have any additional questions, we will be happy to answer them!

---

### Meta-Review · Area_Chair_77fL · 2024-12-23

**Metareview:**

The paper introduces GenieWorksheets, a framework for developing task-oriented conversational agents that integrate task management and knowledge access. By providing a declarative language for defining dialogue tasks, API interactions, and knowledge integration, Genie aims to enhance the reliability and scalability of LLM-based agents. Experimental results demonstrate performance improvements over GPT-4 Turbo on various benchmarks, including STARv2, with better execution accuracy and goal completion rates.

Despite these contributions, the paper has critical weaknesses. The novelty of GenieWorksheets is limited, as it primarily integrates existing ideas in a structured way without substantial innovation. Comparisons with established frameworks like LangChain and AutoGen are inadequate, and the evaluation lacks diverse tasks and datasets to validate generalizability. Methodological clarity is also an issue, with insufficient details provided for reproducibility. The presentation is difficult to follow, with high-level abstractions and limited concrete examples, reducing the paper’s accessibility.

Strengths include the focus on reliability and the attempt to bridge task-oriented and knowledge-based functionalities in conversational agents. However, the lack of strong experimental validation, insufficient novelty, and poor presentation outweigh the contributions.

**Additional Comments On Reviewer Discussion:**

Reviewers noted the framework’s potential but criticized the limited innovation, insufficient baseline comparisons, and unclear methodology. While the authors provided additional experiments and clarified some points in the rebuttal, key issues, such as reproducibility and generalizability, remain unresolved. The framework’s reliance on specific LLM capabilities and its limited scope of evaluation were particularly problematic. The rebuttal efforts improved understanding but did not fully address these concerns, emphasizing the need for a more comprehensive revision in the future.

---

### Decision · Program_Chairs · 2025-01-22

Reject